# STEPWISE GUIDED POLICY OPTIMIZATION: COLORING YOUR INCORRECT REASONING IN GRPO

## ABSTRACT

Reinforcement learning (RL) has proven effective in strengthening the reasoning capabilities of large language models (LLMs). A widely adopted method, Group Relative Policy Optimization (GRPO) (Shao et al., 2024), has shown strong empirical results in training DeepSeek-R1 (Guo et al., 2025a). However, GRPO fails to update the policy when all responses within a group are incorrect (i.e., *all-negative-sample* groups). This limitation underscores a key gap between artificial and human intelligence: unlike humans, who can learn from mistakes, GRPO discards these signals. Our first contribution is to introduce a simple framework that mitigates the all-negative-sample issue by incorporating response diversity within groups using a *step-wise* judge model, which can be either directly trained or adapted from existing LLMs. We prove that this diversification can accelerate GRPO's learning dynamics in a simplified setting. We also empirically validate the proposed stepwise guided policy optimization (SGPO) method, demonstrating consistent gains across model sizes (7B, 14B, 32B) in offline and online training on 9 benchmarks, including base and distilled variants. Our results highlight two advantages: (i) SGPO surpasses GRPO, especially in the early and mid-training stages where all-negative-sample groups are prevalent; and (ii) SGPO does not require judge models to generate correct answers, differentiating it from knowledge distillation methods.

## 1 INTRODUCTION

The rise of OpenAI-o1 (Jaech et al., 2024), DeepSeek-R1 (Guo et al., 2025a), and Kimi-1.5 (Team et al., 2025) has highlighted the emergence of *large AI reasoning models*. Unlike instruction-tuned models (Brown et al., 2020; Chowdhery et al., 2023; Touvron et al., 2023; Achiam et al., 2023), which produce quick responses by statistically inferring the next token, these new reasoning models deliberately decompose complex prompts (e.g., mathematical problems) into intermediate steps and work through chain-of-thought reasoning (Wei et al., 2022; Yao et al., 2023; Besta et al., 2024; Xiang et al., 2025). This slower yet more rigorous process yields greater accuracy and makes them more human-like, enabling success on more complex and challenging tasks (Yang et al., 2018; Shi et al., 2024; Jain et al., 2025). As the generative AI applications move beyond simple conversational interfaces, these reasoning models are poised to become increasingly powerful and widely adopted, positioning them as a key frontier in practice.

At the heart of this revolution lies post-training with outcome-based and verifiable rewards (Cobbe et al., 2021; Uesato et al., 2022; Zelikman et al., 2022; Singh et al., 2023; Hosseini et al., 2024; Lightman et al., 2024; Wang et al., 2024; Setlur et al., 2025; Zhang et al., 2025b), together with reinforcement learning (RL) methods (Schulman et al., 2015; 2017; Li et al., 2024b; Ahmadian et al., 2024; Shao et al., 2024; Xiong et al., 2025a), appreciated for their simplicity, intuitiveness, and practicality. A leading approach is proximal policy optimization (PPO) (Schulman et al., 2017), which relies on a critic (or value) model to estimate advantages. While essential in general RL tasks, this critic is often unnecessary in large language models (LLMs) due to their deterministic transition dynamics (Li et al., 2024b). This observation has inspired alternatives such as group relative policy optimization (GRPO) (Shao et al., 2024) and its extensions (Yu et al., 2025b; Liu et al., 2025b; Chu et al., 2025; Zhang et al., 2025a), which estimate advantages directly in a group-relative fashion.

A major limitation of these methods arises when all sampled responses in a group are incorrect (i.e., *all-negative-sample* groups), which eliminates the learning signal and halts policy updates. In GRPO, given a prompt $\mathbf{x}$, responses $\{\mathbf{y}_i\}_{i=1}^{G}$ are drawn from the old policy $\pi_{\text{old}}$ and assigned rewards $\{r_i\}_{i=1}^{G}$, where $r_i = 1$ if $\mathbf{y}_i$ is correct and 0 otherwise. Advantages are obtained by normalizing $r_i$ within the group. If $r_i = 0$ for all $i$, the advantage vanishes, yielding no update. Such groups are frequent in early and mid-stages of training, when reasoning ability is weak[1]. This shortcoming highlights a gap between artificial and human intelligence: humans effectively learn from mistakes, which act as essential signals during cognitive development (Chialvo & Bak, 1999). In mathematical reasoning, all-negative-sample groups prompt a child to revise rules and strengthen reasoning ability.

Recent studies suggest that negative samples in RL-based large reasoning model training carry more nuanced value than previously assumed (Xiong et al., 2025a). Instead of treating negative samples uniformly, they advocate for principled mechanisms to distinguish negative samples. One prominent direction is process reward models (PRMs) (Lightman et al., 2024; Wang et al., 2024; Luo et al., 2024; Setlur et al., 2025; Zhang et al., 2025b), which estimate either the probability of final success or its change after each reasoning step. However, their reliance on speculative value functions makes them prone to reward hacking (Skalse et al., 2022).

The key insight is that many reasoning tasks possess a structure where step-level correctness can be explicitly defined. This motivates the use of a step-wise judge model that evaluates trajectories by labeling each step as correct (1) or incorrect (0). Such a model can be trained directly (Xiong et al., 2025b) or adapted from existing LLMs (Zha et al., 2025; He et al., 2025)[2]. By grounding rewards in step-level correctness rather than speculative value estimates, our method mitigates reward hacking and yields clearer signals. Intuitively, this allows negative samples to be differentiated through their trajectories: while early-stage reasoning trajectories are of low-quality, these remain informative – much like partial credit in education, where intermediate steps still guide learning.

Our approach enables a holistic evaluation of multi-step reasoning by transforming negative samples from binary outcome rewards into graded, step-level rewards. Consider a negative sample with five reasoning steps $(a_1, a_2, a_3, a_4, a_5)$. If the first error occurs at $a_3$, then $a_1$ and $a_2$ are correct, yielding a correctness proportion of $\frac{2}{5}$. To improve reliability, we adopt a Grok4-Heavy -inspired strategy where multiple independent judgments are obtained from the judge model, and the error position is determined by the majority vote. We further introduce two scaling parameters $\beta$ and $\gamma$ to downweight noisy or unreliable signals (see Eq. (2)). Unlike PRMs, our approach avoids memory overhead and does not require costly step-level human annotations, thereby accelerating training.

**Contribution.** We propose and analyze a simple and efficient framework that introduces response diversity within all-negative-sample groups. It is both theoretically grounded in the simplified setting and empirically effective on various models, distinguishing our approach from existing heuristics. Our contributions can be summarized as follows:

1. We propose a *Stepwise Guided Policy Optimization* (SGPO) framework that leverages a step-wise judge model that identifies the first incorrect step that causes the trajectory to deviate from correctness. This makes evaluation computationally tractable and reliable. *It is important to emphasize that our contribution lies not in designing effective judge models, but in introducing a framework that leverages step-wise judges to effectively distinguish negative samples.* We also prove that SGPO outperforms GRPO in a simplified setting.

2. We conduct the experiments demonstrating the effectiveness of our approach in improving LLM reasoning. Evaluations are undertaken across various model sizes (7B, 14B, 32B) in both offline and online settings with nine benchmarks, including base and distilled variants. Our results reveal two key benefits: (i) SGPO delivers improvements beyond the reach of GRPO, especially in the early and mid-stages of training where all-negative-sample groups are common; (ii) SGPO does not rely on more powerful judge models generating correct answers, allowing it to be distinguish from knowledge distillation methods.

---

[1]To reduce computational cost, training often uses small group sizes and short rollouts, further increasing the likelihood of all-negative-sample groups.

[2]We do not have access to their judge models as it's not publicly released, so we adapt our own from existing LLMs.

The additional overhead from all-negative-sample groups remains modest, since the correctness can be efficiently verified against reference solutions, enabling rapid assessment of reasoning steps. As the computational and financial costs of closed-source judge models (`o4-mini`, `Claude3.7`) rise, SGPO accelerates learning dynamics, making the trade-off worthwhile. SGPO also outperforms GRPO with less powerful and more affordable open-source judge models (`DeepSeek-V3-0324`, `Qwen3-235B-A22B`, `QwQ-32B`), confirming that SGPO remains effective even without cutting-edge LLMs and underscoring its practicality in lower-resource settings.

## 2 PRELIMINARIES AND TECHNICAL BACKGROUND

Modern LLMs are built based on the Transformer architecture (Vaswani et al., 2017) and generate responses $\mathbf{y} = (a_1, \ldots, a_H)$ to user prompts $\mathbf{x}$, where each token $a_h \in \mathcal{V}^\star$, with $\mathcal{V}$ denoting the vocabulary and $\mathcal{V}^\star$ the set of all possible token sequences. We view the LLM as a policy $\pi_\theta(\mathbf{y}|\mathbf{x})$ parameterized by $\theta$, assigning probabilities to responses $\mathbf{y}$ given $\mathbf{x}$. The policy operates in an auto-regressive way as follows:

$$\pi_\theta(\mathbf{y}|\mathbf{x}) = \prod_{h=1}^{H} \pi_\theta(a_h \mid \mathbf{x}, a_1, \ldots, a_{h-1}).$$

For a prompt $\mathbf{x}$ with ground-truth response $\mathbf{y}_\mathbf{x}^\star$, performance is evaluated using a regular-expression match on the final answer: $r(\mathbf{x}, \mathbf{y}) = 1$ if $\mathbf{y}$ matches $\mathbf{y}_\mathbf{x}^\star$ and $r(\mathbf{x}, \mathbf{y}) = 0$ otherwise (Hendrycks et al., 2021). We consider the reasoning tasks defined over a dataset $\mathcal{D} = (\mathbf{x}, \mathbf{y}_\mathbf{x}^\star)$, where each $\mathbf{x}$ is a problem and $\mathbf{y}_\mathbf{x}^\star$ its ground-truth solution.

The policy gradient methods (Williams, 1992; Sutton & Barto, 1998) aim to maximize the objective $J(\theta) = \mathbb{E}_{\mathbf{x} \sim \rho, \mathbf{y} \sim \pi_\theta(\cdot|\mathbf{x})}[r(\mathbf{x}, \mathbf{y})]$ where $\rho$ is the prompt distribution and $\pi_\theta$ is an LLM policy. Parameters are updated via $\theta \leftarrow \theta + \eta \nabla_\theta J(\theta)$. In practice, trajectories are sampled from an old policy $\pi_{\theta_{\text{old}}}$, which is different from $\pi_\theta$, motivating the use of importance sampling as follows:

$$J(\theta) = \mathbb{E}_{\mathbf{x} \sim \rho, \mathbf{y} \sim \pi_{\theta_{\text{old}}}(\cdot|\mathbf{x})} \left[ \frac{\pi_\theta(\mathbf{y}|\mathbf{x})}{\pi_{\theta_{\text{old}}}(\mathbf{y}|\mathbf{x})} r(\mathbf{x}, \mathbf{y}) \right].$$

However, this estimator suffers from high variance when $\pi_\theta$ deviates from $\pi_{\theta_{\text{old}}}$. To stabilize training, clipped surrogate objectives are used as follows:

$$J(\theta) = \mathbb{E}_{\mathbf{x} \sim \rho, \mathbf{y} \sim \pi_{\theta_{\text{old}}}(\cdot|\mathbf{x})} \left[ \min \left\{ \frac{\pi_\theta(\mathbf{y}|\mathbf{x})}{\pi_{\theta_{\text{old}}}(\mathbf{y}|\mathbf{x})} r(\mathbf{x}, \mathbf{y}), \texttt{clip} \left\{ \frac{\pi_\theta(\mathbf{y}|\mathbf{x})}{\pi_{\theta_{\text{old}}}(\mathbf{y}|\mathbf{x})}, 1 - \epsilon, 1 + \epsilon \right\} r(\mathbf{x}, \mathbf{y}) \right\} \right].$$

The group relative policy optimization (GRPO) and its variants (Yu et al., 2025b; Liu et al., 2025b; Chu et al., 2025; Zhang et al., 2025a) adopt this framework but estimate gradients using groups of samples. For each prompt $\mathbf{x}$, GRPO samples responses $\mathbf{y}_1, \ldots, \mathbf{y}_G$ from $\pi_{\theta_{\text{old}}}$ and maximizes the objective function in the form of

$$J(\theta) = \frac{1}{G} \sum_{i=1}^{G} \left[ \min \left\{ \frac{\pi_\theta(\mathbf{y}_i|\mathbf{x})}{\pi_{\theta_{\text{old}}}(\mathbf{y}_i|\mathbf{x})} A_i, \texttt{clip} \left\{ \frac{\pi_\theta(\mathbf{y}_i|\mathbf{x})}{\pi_{\theta_{\text{old}}}(\mathbf{y}_i|\mathbf{x})}, 1 - \epsilon, 1 + \epsilon \right\} A_i \right\} \right],$$

where $\epsilon \in (0, 1)$ and the advantage $A_i$ is computed as

$$A_i = \frac{r(\mathbf{x}, \mathbf{y}_i) - \texttt{mean}(\{r(\mathbf{x}, \mathbf{y}_1), \ldots, r(\mathbf{x}, \mathbf{y}_G)\})}{\texttt{std}(\{r(\mathbf{x}, \mathbf{y}_1), \ldots, r(\mathbf{x}, \mathbf{y}_G)\})}, \tag{1}$$

where $r(\mathbf{x}, \mathbf{y}_i) = 1$ if $\mathbf{y}_i$ matches the ground-truth answer and 0 otherwise.

**Remark 2.1.** *When rewards are identical across all samples within a group, $A_i = 0$ and no update occurs. This is appropriate for all-positive groups but constitutes a critical limitation for all-negative groups, where GRPO fails to exploit mistakes as learning signals.*

## 3 MAIN RESULTS

We propose the Stepwise Guided Policy Optimization (SGPO) framework, which employs the stepwise judge model to detect the first incorrect step that leads a trajectory away from correctness. In a simplified setting, we prove that SGPO consistently accelerates GRPO's learning dynamics.

### 3.1 A STEP-WISE JUDGE MODEL

We propose a principled reward mechanism for negative samples, wherein the step-wise judge model differentiates between structurally sound but partially incorrect reasoning and entirely erroneous responses. This design is motivated by the intuition that an incorrect final answer does not invalidate the entire reasoning process. For instance, a model may follow a logically coherent sequence of steps yet make a minor error – such as an arithmetic slip – that leads to an incorrect conclusion. Treating such cases the same as fundamentally flawed or incoherent reasoning does not make sense. This refinement remains effective under constraints such as reduced output length, where a model may be unable to complete the full solution but still demonstrates a valid reasoning trajectory.

Our step-wise judge model can be either trained directly or adapted from existing LLMs. It evaluates responses sequentially, identifying the first substantive error – such as a computational slip or a logical fallacy – that causes the trajectory to deviate from correctness. To formalize this, we define the *Reasoning Trajectory Score* (RTS) for an incorrect response $\mathbf{y}$, denoted as $\mathrm{RTS}(\mathbf{y}) \in [0, 1]$. The judge model checks each step in order, pinpoints the first error, and treats all preceding steps as the valid reasoning segment. $\mathrm{RTS}(\mathbf{y})$ is then computed as the ratio of the valid segment length to the total trajectory length. For example, if $\mathbf{y}$ consists of five steps $(a_1, a_2, a_3, a_4, a_5)$ and the first error occurs at $a_4$, then $\mathrm{RTS}(\mathbf{y}) = \frac{3}{5}$, indicating that three steps of reasoning are correct before erroneous.

In our experiment, we adapt the judge model from existing LLMs, either closed-source (`o4-mini`, `Claude3.7`) or open-source (`DeepSeek-V3-0324`, `Qwen3-235B-A22B`, `QwQ-32B`). To enhance reliability and further reduce variance in the reward signal, we employ the following protocol: (i) alongside the candidate response, we provide a reference solution drawn from a supervised fine-tuning dataset with correct answers and reasoning trajectories, anchoring the intended solution path and enabling error localization; and (ii) we elicit step-wise evaluation rather than holistic evaluation. The judge model justifies correctness or flags an error sentence by sentence, identifies the first clear mistake, and then traces how this error propagates to the final incorrect conclusion.

Based on the reasoning trajectory score, we introduce a new outcome reward function:

$$r_{\text{SGPO}}(\mathbf{y}) = \begin{cases} 1, & \text{if the final answer of } \mathbf{y} \text{ is correct,} \\ \frac{1}{1+\exp(\beta(\mathrm{RTS}(\mathbf{y})-\gamma))}, & \text{otherwise.} \end{cases} \tag{2}$$

where $\gamma > 0$ and $\beta > 0$ are two parameters to decide scale threshold and scale intensity, respectively. This design ensures that the model receives a more informative gradient signal during training, thereby encouraging refinement of partially correct reasoning rather than indiscriminate penalization of all incorrect outputs. This specification of $r_{\text{SGPO}}$ can be directly incorporated into the advantage calculation in Eq. (1). As a consequence, we refer to SGPO as GRPO using Eq. (2).

**Remark 3.1.** *Our approach differs from process reward models (PRMs) (e.g. Lightman et al., 2024). For a prompt $\mathbf{x}$ and a prefix of reasoning steps $(a_1, \ldots, a_t)$, a PRM typically predicts either (i) a prefix-level value $V(\mathbf{x}, a_{1:t}) = \mathbb{P}(\text{final answer correct} \mid \mathbf{x}, a_{1:t})$, or (ii) a step-level progress signal such as $\Delta_t = V(\mathbf{x}, a_{1:t}) - V(\mathbf{x}, a_{1:t-1})$. In practice, PRMs are trained by supervised ranking of intermediate steps and are used to re-rank trajectories or shape training at the* prefix *level, acting as approximate value (or Q-value) functions over prefixes. In contrast, SGPO introduces a different way of producing and using feedback signals: (i) Policy-guided rollouts without search. All trajectories are sampled from the current policy, without PRM-guided exploration or trajectory alteration; (ii) Post-hoc first-error identification. A step-wise judge inspects the entire trajectory, pinpoints the earliest error relative to a reference solution, and converts this into a calibrated scalar reward $r_{\text{SGPO}}(\mathbf{y})$ via the reasoning trajectory score; (iii) Stable credit assignment in all-negative-sample groups. By locating the first definitive mistake only after observing the full trace, SGPO eliminates the look-ahead ambiguity and feedback loops inherent to PRM-guided search (Zhang et al., 2024a), while avoiding the need for the judge to solve the problem or approximate a value function.*

**Remark 3.2.** *Our approach differs from knowledge distillation (e.g. Kang et al., 2023; Gu et al., 2024). The student model trained via knowledge distillation inherits the judge model's failure, since it only imitates the judge model's outputs. For instance, consider the AIME problem: "The twelve letters $\{A, B, C, D, E, F, G, H, I, J, K, L\}$ are randomly grouped into six pairs. Each pair is ordered alphabetically to form a two-letter word, and the six words are listed alphabetically, such as $AB, CD, EF, GH, IJ, KL$. The probability that the last word listed contains $G$ is $\frac{m}{n}$ with $m, n$ coprime. Find $m + n$". Neither the student model (`DeepSeek-R1-Distill-Qwen-7B`) nor the judge models (`DeepSeek-V3-0324`) can solve this problem within 16 rollouts. In contrast,*

*SGPO leverages the judge model to identify mistakes in the student's reasoning, providing learning signals that go beyond imitation and enabling improvements unattainable by knowledge distillation.*

### 3.2 ACCELERATING LEARNING DYNAMICS

We present a theoretical analysis to explain why SGPO outperforms GRPO. To this end, we consider a simplified setting with a reasoning horizon of $H = 2$, where each step admits two possible actions $a_h \in 1, 2$ for $h = 1, 2$. This configuration follows prior works (Dayan, 1991; Li et al., 2024b), where analogous examples were employed to validate theoretical insights. Without loss of generality, we assume a unique ground-truth response $y_x^\star = (2, 2)$ for the prompt $x$. The algorithm iteratively updates the policy parameter $\theta$ using samples drawn from the current policy $\pi_\theta$. For clarity, we restrict the sample space to $(1, 1), (2, 1), (2, 2)$, excluding $(1, 2)$ since a correct reasoning step is unlikely to, and should not, follow an incorrect precursor.

To illustrate the effect of SPO, we analyze the learning dynamics of SGPO and GRPO in this simplified setting. Under GRPO, the rewards are assigned as $r((2, 2)) = 1$ and $r((2, 1)) = r((1, 1)) = 0$, meaning that only selecting the "good" action 2 at both steps yields a positive reward. In contrast, SGPO assigns $r_{\text{SGPO}}((2, 2)) = 1$, $r_{\text{SGPO}}((2, 1)) = \frac{1}{2}$ and $r_{\text{SGPO}}((1, 1)) = 0$. The difference is that partial progress – choosing the "good" action 2 in the first step but failing at the second – receives no credit in GRPO yet proportional credit in SGPO. Here, $\frac{1}{2}$ is chosen for illustrative purposes to convey the qualitative behavior of the reward mechanism, while the exact values used in experiments are determined by Eq. (2).

In our analysis, we examine the population-level learning dynamics with $G = 2$, omitting clipping and importance sampling. Let $p_{\text{GRPO}}^{(k)}$ and $q_{\text{GRPO}}^{(k)}$ denote the probability of selecting the "good" action in the first step at iteration $k$ under GRPO, and that of selecting the "good" action in the second step conditioned on a correct first step. Analogously, $p_{\text{SGPO}}^{(k)}$ and $q_{\text{SGPO}}^{(k)}$ denote the corresponding probabilities under SGPO. Our theoretical findings are summarized in the following theorem.

**Theorem 3.3.** *Suppose that $p_{\text{GRPO}}^{(0)} = q_{\text{GRPO}}^{(0)} = p_{\text{SPO}}^{(0)} = q_{\text{SPO}}^{(0)} = \frac{1}{2}$ and $\eta = 1$[3] for GRPO and SGPO. Then, we have that (i) GRPO and SGPO achieve successful learning: $p_{\text{GRPO}}^{(k)}, q_{\text{GRPO}}^{(k)}, p_{\text{SGPO}}^{(k)}, q_{\text{SGPO}}^{(k)} \to 1$ as $k \to +\infty$; (ii) SGPO outperforms GRPO in learning the "good" action in the first step: $p_{\text{SGPO}}^{(k)} > p_{\text{GRPO}}^{(k)}$ for all $k \geq 1$; (iii) SGPO outperforms GRPO in learning the optimal policy: $p_{\text{SGPO}}^{(k)} q_{\text{SGPO}}^{(k)} > p_{\text{GRPO}}^{(k)} q_{\text{GRPO}}^{(k)}$ for all $k \geq 1$.*

*Proof Sketch.* For (i), we first show the sequence $(p_{\text{SGPO}}^{(k)})_{k \geq 1}$ is strictly increasing and bounded in $(0, 1)$ (see Lemmas C.4(i) and C.4(ii)). Thus, it converge to some value $c \in (0, 1]$. Take limit as $k \to \infty$ on both sides of the update rule, we obtain the only feasible solution $c = 1$. Similarly, we show $q_{\text{SGPO}}^{(k)}, p_{\text{GRPO}}^{(k)}, q_{\text{GRPO}}^{(k)} \to 1$ as $k \to \infty$. For (ii), we use induction. For the base case, we use the update rule and Lemma C.1(ii). Assume $p_{\text{SGPO}}^{(k)} > p_{\text{GRPO}}^{(k)}$ for some $k \geq 1$, we derive $p_{\text{SGPO}}^{(k+1)} > p_{\text{GRPO}}^{(k+1)}$ using Lemmas C.1(i) and C.1(ii). For (iii), we show $p_{\text{GRPO}}^{(k)} = q_{\text{GRPO}}^{(k)}$ for all $k \geq 1$ using induction. It suffices to show $p_{\text{SGPO}}^{(k)} q_{\text{SGPO}}^{(k)} > (p_{\text{GRPO}}^{(k)})^2$ for all $k \geq 1$. We prove using induction again. For the base case, we use Lemma C.1(iv). It remains to show that

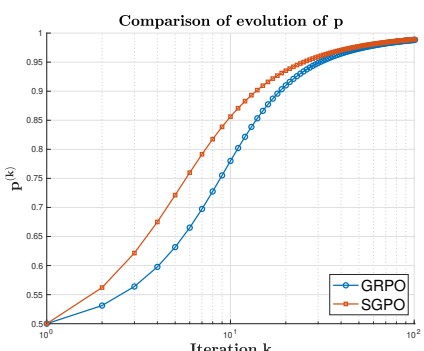

Figure 1: SGPO outperforms GRPO in learning a "good" action in the first step.

$p_{\text{SGPO}}^{(k)} q_{\text{SGPO}}^{(k)} > (p_{\text{GRPO}}^{(k)})^2$ implies $p_{\text{SGPO}}^{(k+1)} q_{\text{SGPO}}^{(k+1)} > (p_{\text{GRPO}}^{(k+1)})^2$. Note that Lemma C.4(iii) guarantees $p_{\text{SGPO}}^{(k)} > q_{\text{SPO}}^{(k)}$ for all $k \geq 1$. Thus, Lemma C.2 implies that

$$p_{\text{SGPO}}^{(k+1)} q_{\text{SGPO}}^{(k+1)} = \frac{1}{A(p_{\text{SGPO}}^{(k)}) B(p_{\text{SGPO}}^{(k)}, q_{\text{SGPO}}^{(k)})} > \frac{1}{C\left(\sqrt{p_{\text{SGPO}}^{(k)} q_{\text{SGPO}}^{(k)}}\right)^2},$$

---

[3]We use the unit stepsize for simplicity. Our results are valid for any sufficiently small step size.

Table 1: Evaluation results on offline RL training. For each model, we report the baseline performance before RL training. We then report RL training results that uses only negative samples and positive samples, respectively. Performance across validation and training dataset (`LIMO`) is shown.

| | AMC23 avg@16 | AIME24 avg@16 | MATH500 pass@1 | Olympiads pass@1 | LIMO pass@1 |
|---|---|---|---|---|---|
| **Qwen2.5-14B-Instruct** | | | | | |
| Baseline | 58.59 | 14.58 | **80.40** | 41.78 | 31.70 |
| Negative Samples only | **61.88** | **15.21** | **80.40** | **42.37** | 30.11 |
| Positive Samples only | 61.72 | 14.58 | 79.80 | 42.07 | **38.68** |
| **Qwen2.5-32B-Instruct** | | | | | |
| Baseline | 64.22 | 17.08 | **83.60** | 45.93 | 34.64 |
| Negative Samples only | **69.53** | **20.42** | 83.00 | 46.37 | 36.47 |
| Positive Samples only | 66.87 | 18.75 | **83.60** | 47.41 | **41.86** |

where the functions $A(\cdot)$, $B(\cdot)$ and $C(\cdot)$ are defined in Lemma C.2. By using Lemma C.1(iii), we complete the induction by applying our induction hypothesis. □

**Remark 3.4.** *Theorem 3.3 presents one of the first theoretical analyses of GRPO with multiple samples and multi-step reasoning in the context of LLM reasoning. The first part establishes that SGPO converges to the optimal policy. The second and third parts demonstrate that SGPO both accelerates the acquisition of partially correct reasoning steps and preserves partial reasoning ability even when the final answer is incorrect. Importantly, the theorem provides a **per-iteration** comparison of learning under different reward mechanisms – an aspect rarely examined in previous works. The provable improvement in learning the optimal policy is also consistent with our numerical findings as partly shown in Figure 1; see Appendix C.4 for further details.*

## 4 EXPERIMENTS

We demonstrate the benefits of differentiating negative samples through experiments in both offline and online settings. Offline RL is more computationally efficient, offering faster training, reduced memory consumption, and improved stability. In contrast, online RL provides greater flexibility and learning capacity, and has become the standard approach in large-scale reasoning models such as DeepSeek-R1 (Guo et al., 2025a).

### 4.1 OFFLINE TRAINING

For baselines, we consider strong models without further fine-tuned on math-specific SFT datasets, namely `Qwen2.5-14B-Instruct` and `Qwen2.5-32B-Instruct`. Prior work showed that a small set of carefully curated prompts significantly enhance the reasoning capability. Accordingly, we adopt the `GAIR/LIMO` dataset (Ye et al., 2025) as the training set, which has demonstrated strong potential for improving the reasoning performance of large-scale (32B) models in offline SFT. Evaluation is conducted on four standard math reasoning benchmarks: `AIME24`, `AMC23`, `MATH500` (Hendrycks et al., 2021), and `OlympiadBench` (He et al., 2024). Our aim is to highlight the rich learning signal contained in all-negative-sample groups, showing that training exclusively on them can still yield performance gains. For benchmarks with fewer than 100 questions (`AMC23`, `AIME24`), we report `avg@16` results with a decoding temperature of 0.6 and `Top_P = 0.95`. For benchmarks with more than 100 questions, we report `pass@1` results using greedy decoding. Across all experiments, the maximum decoding length is set to 32768 tokens.

We conduct all response generation and model updates using offline RL (Peters & Schaal, 2007) with the standard GRPO mechanism. Specifically, the model is updated with advantages estimated from the offline dataset (see, e.g., Peng et al., 2019; Li et al., 2024b). For each prompt, we sample six responses per group and identify all-negative-sample groups in which all responses yield incorrect answers. Within these groups, we apply the step-wise judge model to assign differentiated rewards to negative samples, which are then used for offline RL updates. The model is trained for three epochs with a learning rate of $2 \times 10^{-6}$. As a contrastive baseline, we also perform offline RL using

Table 2: Evaluation results on online RL training. We refer to BASELINE as the performance of the original model without RL finetuning. **Overall** is average performance across all the benchmarks. Note that the training dataset is `AIME1997-2023`. For `DeepSeek-R1-Distill-Qwen-7B`, we report additional results, including (i) compatibility with more judge models and (ii) ablation on the stability parameters $\beta$ and $\gamma$.

| | Kaoyan pass@1 | GradeMath pass@1 | MATH500 pass@1 | Olympiads pass@1 | CHMath24 avg@16 | AIME25 avg@16 | AIME24 avg@16 | GaoKao avg@16 | AMC23 avg@16 | Overall avg |
|---|---|---|---|---|---|---|---|---|---|---|
| **DeepSeek-R1-Distill-Qwen-7B** | | | | | | | | | | |
| BASELINE | 50.25 | 41.43 | 87.00 | 49.93 | 73.75 | **40.62** | 52.92 | 80.22 | 89.53 | 62.85 |
| GRPO | 55.78 | 43.33 | 89.40 | **56.00** | 71.04 | 36.68 | 52.08 | 80.30 | 88.91 | 63.72 |
| SGPO+o4-mini-0416 | 57.79 | 46.19 | 90.80 | 54.67 | 75.00 | 38.33 | 54.58 | 81.33 | 90.00 | 65.41 |
| SGPO+DeepSeek-V3-0324 | 54.77 | **47.17** | 91.00 | 55.11 | **77.29** | 40.42 | **56.87** | 82.28 | 90.83 | **66.19** |
| SGPO+Qwen3-235B-A22B | 56.78 | 46.67 | 92.00 | 54.67 | 73.33 | 37.92 | 55.63 | 81.17 | 90.63 | 65.42 |
| SGPO+QwQ-32B | 52.26 | 45.24 | 92.00 | 53.78 | 75.00 | 35.21 | 56.46 | 82.28 | 91.88 | 64.91 |
| SGPO+QwQ-32B w/o {$\beta,\gamma$} | **58.29** | 42.38 | 90.20 | 55.11 | 74.58 | 38.69 | 53.63 | 81.24 | 88.75 | 65.08 |
| **DeepSeek-R1-Distill-Llama-8B** | | | | | | | | | | |
| BASELINE | 29.15 | 23.81 | 77.40 | 41.48 | **61.46** | 27.92 | **42.29** | **72.78** | 87.97 | 51.58 |
| GRPO | 35.68 | 28.33 | 84.00 | 46.32 | 57.08 | **28.33** | 42.08 | 68.99 | 86.72 | 53.06 |
| SGPO+Claude-3.7 | **39.70** | **29.05** | 83.60 | **48.44** | 58.96 | 24.58 | 39.37 | 71.52 | **89.06** | **53.81** |
| **Qwen2.5-14B-Instruct** | | | | | | | | | | |
| BASELINE | 37.69 | 49.52 | 80.40 | 41.78 | 21.88 | 13.13 | **14.58** | **41.14** | 58.59 | 39.85 |
| GRPO | **43.22** | 47.14 | 80.20 | 43.11 | 21.88 | 13.13 | 13.33 | 39.16 | **59.84** | 40.11 |
| SGPO+o4-mini-0416 | 38.69 | **53.33** | **81.00** | **44.00** | **22.92** | **16.67** | 14.17 | 39.00 | 59.22 | **41.00** |
| **Qwen2.5-32B-Instruct** | | | | | | | | | | |
| BASELINE | 45.73 | **53.81** | **83.60** | 45.93 | 26.87 | 12.29 | 17.08 | 44.15 | 64.22 | 43.74 |
| GRPO | **48.24** | 52.86 | 83.20 | 45.93 | 22.50 | 12.08 | **21.67** | 45.73 | 67.34 | 44.39 |
| SGPO+o4-mini-0416 | **48.24** | **53.81** | 83.00 | **46.81** | **29.79** | **14.58** | 19.58 | 45.09 | **69.53** | **45.06** |
| **QwQ-32B** | | | | | | | | | | |
| BASELINE | 64.32 | 62.38 | 94.60 | 68.74 | **89.39** | **68.54** | 77.71 | 86.88 | 97.03 | 78.84 |
| GRPO | 71.36 | 63.81 | 94.60 | 69.48 | 88.75 | 64.38 | 75.83 | **87.11** | 97.03 | 79.15 |
| SGPO+DeepSeek-V3-0324 | **73.37** | **64.76** | **95.00** | **70.22** | 88.33 | 66.46 | **78.33** | **87.11** | **97.97** | **80.17** |

only positive rollouts with correct answers. This parallel setup enables a direct comparison between learning from exclusively negative reasoning trajectories and from exclusively positive ones.

We conduct offline RL training to demonstrate that utilizing all-negative-sample groups can enhance the reasoning abilities of LLMs. For comparison, we also include positive-only offline RL training. As shown in Table 1, SGPO with negative samples consistently improve performance across most of benchmarks, in some cases even surpassing models trained solely on positive samples. Notably, in the 14B model experiment, training on negative samples yields improvements on four benchmarks relative to the positive-sample baseline. These findings underscore the utility of negative samples, which should not be discarded in online GRPO training; see more discussions in Section 4.4.

## 4.2 ONLINE TRAINING

For baselines, we consider applying `Qwen2.5-14B-Instruct`, `Qwen2.5-32B-Instruct`, `QwQ-32B`, `DeepSeek-R1-Distill-Qwen-7B` and `DeepSeek-R1-Distill-Llama-8B`. Online GRPO training is implemented using the `verl` framework (Sheng et al., 2025). For the step-wise judge model, we adopt a diverse set of LLMs, ranging from closed-source models with strong reasoning capabilities (`o4-mini`, `Claude3.7`) to open-source models that are more accessible to the community, including `DeepSeek-V3-0324`, `Qwen3-235B-A22B`, and `QwQ-32B`.

Compared to offline RL, online RL yields larger improvements in a model's reasoning capabilities. Since our baselines already include strong distillation models, some benchmarks used in offline evaluation are nearing saturation. To provide a better assessment, we expand our evaluation suite beyond `AMC23`, `AIME24`, `MATH500`, and `OlympiadBench` by including `AIME25`, `GradeSchool` (Ye et al., 2025), `CHMath24`, `Kaoyan`, and `Gaokao`. Specifically, `CHMath24` is the benchmark from the 2024 Chinese High School Mathematics League Competition, `Gaokao` from China's 2024 National College Entrance Examination, `Kaoyan` from the Chinese Graduate School Entrance Examinations, and `GradeSchool` targets elementary-level mathematical reasoning. Among these, `CHMath24` and `Gaokao` each contain fewer than 100 questions, for which we apply the temperature-based decoding for evaluation.

For GRPO training, we use the `AIME` collections from 1997 to 2023 provided in DeepScaler (Luo et al., 2025b), training for 12 epochs. All training questions are in English, while evaluation benchmarks include multilingual questions. Notably, negative samples learned during training generalize

Table 3: Evaluation results are reported for `DeepSeek-R1-Distill-Qwen-7B` across four independent runs. First column indicates judge models and its corresponding reward stability setup.

| | Kaoyan pass@1 | GradeMath pass@1 | MATH500 pass@1 | Olympiads pass@1 | CHMath24 avg@16 | AIME25 avg@16 | AIME24 avg@16 | GaoKao avg@16 | AMC23 avg@16 | Overall avg |
|---|---|---|---|---|---|---|---|---|---|---|
| **DeepSeek-R1-Distill-Qwen-7B-SGPO** | | | | | | | | | | |
| +Qwen3-235B-A22B | 53.90 ± 2.10 | 46.55 ± 0.24 | 91.30 ± 0.87 | 53.45 ± 1.35 | 74.48 ± 1.10 | 37.40 ± 1.05 | 55.73 ± 1.39 | 81.33 ± 0.18 | 90.19 ± 0.48 | 64.92 ± 0.37 |
| +QwQ-32B | 53.89 ± 1.66 | 44.88 ± 1.36 | 91.15 ± 0.84 | 53.71 ± 0.74 | 74.33 ± 1.29 | 37.03 ± 1.23 | 54.76 ± 1.87 | 81.91 ± 0.53 | 90.08 ± 1.23 | 64.64 ± 0.41 |
| +QwQ-32B w/o $\{\beta,\gamma\}$ | 56.14 ± 2.76 | 44.53 ± 2.41 | 90.10 ± 0.66 | 53.64 ± 1.08 | 73.89 ± 1.13 | 38.70 ± 1.80 | 53.63 ± 2.15 | 81.24 ± 0.50 | 88.83 ± 0.81 | 64.52 ± 0.57 |

Table 4: Evaluation results are reported for `DeepSeek-R1-Distill-Qwen-7B` as base model and `QwQ-32B` as judge model with and without majority voting.

| | Kaoyan pass@1 | GradeMath pass@1 | MATH500 pass@1 | Olympiads pass@1 | CHMath24 avg@16 | AIME25 avg@16 | AIME24 avg@16 | GaoKao avg@16 | AMC23 avg@16 | Overall avg |
|---|---|---|---|---|---|---|---|---|---|---|
| **DeepSeek-R1-Distill-Qwen-7B-SGPO** | | | | | | | | | | |
| +QwQ-32B | 53.89 ± 1.66 | 44.88 ± 1.36 | 91.15 ± 0.84 | 53.71 ± 0.74 | 74.33 ± 1.29 | 37.03 ± 1.23 | 54.76 ± 1.87 | 81.91 ± 0.53 | 90.08 ± 1.23 | 64.64 ± 0.41 |
| +QwQ-32B with voting | 56.66 ± 1.66 | 45.24 ± 2.16 | 91.35 ± 0.50 | 53.82 ± 0.94 | 74.19 ± 0.92 | 37.35 ± 2.25 | 55.81 ± 0.99 | 82.12 ± 0.43 | 90.53 ± 1.07 | 65.23 ± 0.18 |

Table 5: Evaluation results are reported in terms of `pass@16` across benchmarks. The first two columns show the total number of questions and the number solved within 16 attempts, while the last two columns report the number of unique questions solved by one method but not the other.

| | SGPO - `pass@16` | GRPO - `pass@16` | SGPO \ GRPO | GRPO \ SGPO |
|---|---|---|---|---|
| `AIME24` | 23/30 | 19/30 | 4 | 0 |
| `AIME25` | 21/30 | 21/30 | 1 | 1 |
| `Gaokao` | 70/79 | 68/79 | 2 | 0 |
| `AMC23` | 39/40 | 38/40 | 1 | 0 |
| `CHMath24` | 27/30 | 25/30 | 2 | 0 |

well to out-of-domain mathematical reasoning tasks. SGPO training follows the same setup. With batch-simultaneous processing, judge model calls take 90 seconds per batch of negatives, adding 10% wall-clock time relative to rollout and update. Step-wise supervision is applied only to all-negative-sample groups during the first three epochs, as we expect this duration to suffice for the model to internalize corrective signals; beyond this point, unresolved examples are more indicative of model capacity limits than learnability. For all models, rollout length is fixed at 8192 tokens and group size at 8. Models less than 8B are trained on 8 H100, 14B models on 16 H100, and 32B models on 32 H200. We adopt the default KL coefficient and learning rate from the `verl` training script (Sheng et al., 2025), and use the LIMO evaluation script (Ye et al., 2025), both of which are standard practices in the community.

A key insight from Table 2 is that stronger models generate higher-quality negative samples, which substantially aid learning. As model capability improves, so does the informativeness of its mistakes. Negative samples broadly fall into two categories: (i) correct reasoning trajectories truncated by output length limits, and (ii) trajectories containing logical errors. The first type remains highly valuable – yet discarded in GRPO – since it preserves meaningful reasoning steps, motivating our step-wise judge model. The second type, though incorrect, still provides informative signals, especially when all samples fail on genuinely challenging problems. Notably, stronger distilled models average 6K tokens per response, compared to only 1K tokens for weaker base models, making truncated but informative negative samples more common in the stronger case. Likewise, their erroneous responses also tend to be richer and more useful for step-level judgment.

### 4.3 OTHER ABLATION STUDIES

To assess reliability of judge models, we evaluate our approach not only with strong closed-source reasoning models but also with publicly available models of weaker capacity: `DeepSeek-V3`, `Qwen3-235B` and `QwQ-32B`. As shown in Table 2 (best-tuned results) and Table 3 (multiple runs with weaker judges), performance remains stable, indicating that weaker judges do not significantly degrade outcomes. We attribute this reliability to two design choices: (i) first-step error identification with a reference answer. SGPO requires the judge only to verify each step against the reference, not to solve the problem, thereby reducing task difficulty and avoiding the pitfalls of generic PRMs; (ii) reward stability parameters $\beta$ and $\gamma$, which filter and smooth noisy signals. As confirmed by

ablations, removing $\beta$ and $\gamma$ increases variance and weakens performance. To improve verification, we incorporate a Grok4−Heavy-inspired strategy: multiple independent evaluations by the judge model, with the error position selected by majority voting. Using `QwQ-32B` as the judge model, `DeepSeek-R1-Distill-Qwen-7B` as the base model, and four rollouts per judgment, we have observed noticeable gains in consistency and stability (see Table 4).

While `avg@16` measures average performance across rollouts, `pass@16` reflects the ability to solve new questions with multiple attempts. As shown in Table 5, SGPO's gains in `pass@16` stem directly from leveraging negative samples. Learning only from solvable problems reinforces existing ability, whereas all-negative-sample groups correspond to genuinely difficult questions where GRPO consistently fails. These are precisely the cases where additional feedback can be most valuable. By providing step-level signals, SGPO rewards near-misses by reinforcing correct reasoning up to the first error, penalizes early failures by discouraging persistent error modes, and exposes blind spots by turning hard cases into informative training signals. In this regard, SGPO provides benefits that GRPO cannot match, covering more hard problems and providing sharper credit assignment, which translates to faster and more reliable learning under realistic compute budgets.

By leveraging richer early-stage signals from negative samples, SGPO can achieve faster and stronger performance than GRPO. As shown in Figure 2, SGPO continues improving beyond epoch 5 by solving several additional hard training problems, whereas GRPO plateaus. This improvement stems from informative negative samples that help resolve previously unsolved problems as also shown in Table 5. We also find that SGPO converges more rapidly to a deterministic policy, as illustrated by the training trajectories in Figure 3 (see Appendix B).

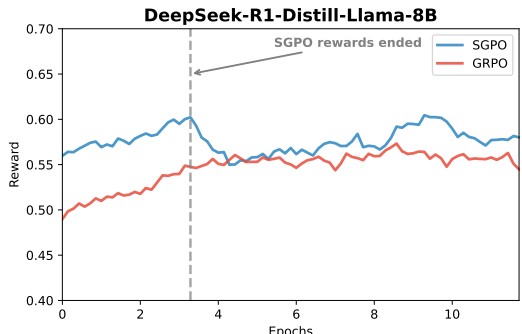

Figure 2: Evaluation results on GRPO and SGPO. SGPO rewards end at epoch 3.

### 4.4 DISCUSSIONS

We highlight the motivation for evaluating both offline and online RL. In the offline setup, training uses **only** negative samples, allowing us to directly test whether incorrect or incomplete reasoning trajectories can improve performance. In the online setup, we simulate realistic GRPO training, where batches contain a random mix of positive and negative samples. This demonstrates that negative samples are not only effective in isolation but also remain valuable in practical settings with noisier, mixed data. While mixing positives and negatives introduces noise, simply discarding negative samples does not stabilize training; in several cases, the performance of GRPO even drops below baseline, as the model overfits to problems it can already solve.

This instability arises from limited out-of-domain generalization and catastrophic forgetting. Without exposure to challenging or partially correct reasoning, the model risks overfitting to easy cases, reinforcing shallow heuristics instead of developing robust problem-solving skills. The absence of diverse failure cases can also cause catastrophic forgetting, degrading performance on previously solvable tasks. Incorporating negative samples mitigates these issues, as SGPO consistently outperforms GRPO on Chinese OOD math benchmarks. Nonetheless, further work is needed to design more stable training frameworks, including richer reward diversification mechanisms for handling negative samples and corresponding efficient RL methods beyond GRPO.

## 5 CONCLUSION

We propose a simple and efficient framework that introduces response diversity within all-negative-sample groups and prove, in a simplified setting, that such diversification can accelerate the learning dynamic of GRPO. Empirically, our approach can yield consistent improvements across model sizes in both offline and online training over nine benchmarks, including base and distilled variants. Future works include extending theoretical results to broader multi-step reasoning tasks, applying response diversity to accelerate other RL methods, and designing lightweight, task-specific reward models that evaluate reasoning steps correctly even if they cannot solve the full problem.

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

# A    RELATED WORKS

We comment on all related topics, including reasoning through test-time compute, chain-of-thought and its variants, direct preference alignment methods, reward models and reinforcement learning from AI feedback. For an overview of more reasoning models and methods, we refer to two recent surveys (Huang & Chang, 2023; Chen et al., 2025c).

**Reasoning through test-time compute.** OpenAI-o1 (Jaech et al., 2024) is among the first large-scale applications of RL to reasoning, and achieved state-of-the-art performance upon release. Following this trend, DeepSeek-R1 (Guo et al., 2025a) is the first open-weight model to match or exceed OpenAI-o1. Their real-world success stories have involved several simple yet novel techniques that enhance LLM reasoning through more test-time compute, including chain-of-thought (Wei et al., 2022), self-consistency (Wang et al., 2023), best-of-$N$ sampling (Snell et al., 2025), process reward models (Lightman et al., 2024), Monte Carlo tree search (Silver et al., 2016; Hao et al., 2023), tree-of-thought (Yao et al., 2023), and recent works on preventing overthinking (Chen et al., 2024b; Team et al., 2025; Luo et al., 2025a; Arora & Zanette, 2025) and compressing chain-of-thought (Hao et al., 2024b; Cheng & Van Durme, 2024). More specifically, *chain-of-thought* is a reasoning approach where intermediate steps are explicitly written to make complex problem-solving processes more transparent and logical. *Self-consistency* suggests generating multiple final answers and returning the mode of an empirical distribution, enhancing test-time performance when test-time verifiers are unavailable. Unfortunately, it is computationally expensive and effective only when answers can be clustered. *Best-of-$N$ sampling* resolves this issue by sampling answers from the model and selecting the best at test time according to the scoring function; however, it is sensitive to the accuracy of test-time scoring functions (Gao et al., 2023). *Process reward models* offer fine-grained supervision of chain-of-thought reasoning, but they might be vulnerable to reward hacking and introduce computation overhead. *Monte Carlo tree search* is a generic technique that allocates computational resources toward the most promising regions of the search space, and *tree-of-thought* and its extension (Besta et al., 2024; Gandhi et al., 2024) simplified this idea by exploring multiple reasoning paths in a specific structure, allowing language models to select the most promising line of thought for complex problem-solving. Both *length regularization* and *compressed chain-of-thought* are developed to reduce inference costs for reasoning, which is crucial for the economic feasibility, user experience and environmental sustainability of LLMs. In addition, several works have focused on specific reasoning tasks (Lampinen et al., 2024; Yang et al., 2025; Srivastava et al., 2024; Huang et al., 2025; 2024; Guo et al., 2025b; Gou et al., 2024; Wang et al., 2025), demonstrating promising performance. The recent findings Xiong et al. (2025a) have shown that the REINFORCE-type methods (including GRPO (Shao et al., 2024)) can not effectively learn from all-negative-sample groups. Our work alleviates this issue by leveraging AI feedback to differentiate negative samples. We also provide a theoretical analysis through a stylized model, explaining why such diversification improves GRPO's learning dynamics.

**Chain-of-Thought and its variants.** Chain-of-thought (CoT) refers to as a broad class of methods that generate an intermediate reasoning process before arriving at a final answer. These approaches either prompt LLMs (Wei et al., 2022; Khot et al., 2023; Zhou et al., 2023) or train LLMs to generate reasoning chains through supervised fine-tuning (SFT) (Yue et al., 2024; Yu et al., 2024b; Li et al., 2025) and/or RL (Wang et al., 2024; Shao et al., 2024; Havrilla et al., 2024; Yu et al., 2025a). While CoT has proven effective for certain tasks, its auto-regressive generation nature makes it challenging to mimic human reasoning on more complex problems (LeCun, 2022; Hao et al., 2023), which require planning and search. Recent efforts were devoted to equipping LLMs with tree search methods (Xie et al., 2023; Yao et al., 2023; Hao et al., 2024a) or training LLMs on search trajectories (Lehnert et al., 2024; Gandhi et al., 2024; Su et al., 2025). Several other works have investigated why CoT is effective. For example, (Madaan et al., 2023) used a counterfactual prompting approach to examine the relative contributions of prompt elements, including symbols (digits, entities) and patterns (equations). (Feng et al., 2023; Merrill & Sabharwal, 2024; Li et al., 2024a) analyzed CoT from the perspective of model expressivity, and (Feng et al., 2023) showed that employing CoT increases the effective depth of a transformer since the generated outputs are looped back to the input. This insight motivated the chain-of-continuous-thought paradigm (Hao et al., 2024b), and a related approach has been proposed in (Cheng & Van Durme, 2024).

**Direct preference alignment methods.** These methods (e.g., DPO (Rafailov et al., 2023)) are simple and stable offline alternatives to online RLHF. Various DPO variants with other objectives

have been proposed, including ranking ones beyond pairwise preference data (Dong et al., 2023; Yuan et al., 2023a; Song et al., 2024; Chen et al., 2024a; Liu et al., 2025a) and simple ones that do not rely on a reference model (Hong et al., 2024; Meng et al., 2024). Since DPO does not train a reward model, the limited size of human labels becomes a bottleneck. To alleviate this limitation, subsequent works proposed to augment preference data using a trained SFT policy (Zhao et al., 2023) or a refined SFT policy with rejection sampling (Liu et al., 2024a). The DPO loss was recently extended to token-level MDP (Rafailov et al., 2024) given that the transition is deterministic – which has covered the fine-tuning of LLMs – and more general RL problems (Azar et al., 2024). There are other DPO variants (Ethayarajh et al., 2024; Park et al., 2024; Xu et al., 2024; Tang et al., 2024; Meng et al., 2024; Chen et al., 2025a; Zhao et al., 2025). For example, (Ethayarajh et al., 2024) designed the specific loss using a prospect theory, (Tang et al., 2024) optimized a general preference loss instead of the log-likelihood loss, and (Meng et al., 2024) aligned the reward function in the preference optimization objective with the generation metric. Dong et al. (2024) and (Xiong et al., 2024) proposed to generate human feedback in an online fashion to mitigate the distribution-shift and over-parameterization phenomenon. This improves DPO for complex reasoning tasks (Pang et al., 2024). Several other works focus on *unintentional alignment* of DPO and developing new methods (Pal et al., 2024; Tajwar et al., 2024; Liu et al., 2024b; Xiao et al., 2024; Yuan et al., 2025; Razin et al., 2025; Chen et al., 2025b). Among these works, (Razin et al., 2025) proposed to measure the similarity between preferred and dispreferred responses using the centered hidden embedding similarity (CHES) score and showed that filtering out preference pairs with small CHES score improves DPO, while (Chen et al., 2025b) proposed to use comparison oracles, and showed that combining it with DPO effectively alleviated the issue of unintentional alignment.

**Reward models.** For the prompt $\mathbf{x}$ with a ground-truth response $\mathbf{y}_{\mathbf{x}}^{\star}$, we evaluate by implementing a regular expression match on the final answer (Hendrycks et al., 2021): $r(\mathbf{x}, \mathbf{y}) = 1$ if $\mathbf{y}$ matches $\mathbf{y}_{\mathbf{x}}^{\star}$ on the *final answer* and $r(\mathbf{x}, \mathbf{y}) = 0$ otherwise. An *outcome reward* model (ORM) (Cobbe et al., 2021; Uesato et al., 2022) is trained for estimating $r(\mathbf{x}, \mathbf{y})$. In particular, we first choose $\mathbf{x} \in \mathcal{D}$ and collect training samples $(\mathbf{x}, \mathbf{y} \sim \pi_{\theta}(\cdot|\mathbf{x}), r(\mathbf{x}, \mathbf{y}))$. Then, we take $(\mathbf{x}, \mathbf{y})$ as input and train an ORM to predict $r(\mathbf{x}, \mathbf{y})$. This can be done using binary classification (Cobbe et al., 2021; Yu et al., 2024a), direct preference optimization (Hosseini et al., 2024) or next-token prediction (Zhang et al., 2024b). Previous works also train LLMs on self-generated data using the ground-truth outcome reward model with either supervised fine-tuning (Singh et al., 2024; Yuan et al., 2023b; Zelikman et al., 2022) or online RL (Bi et al., 2024; Guo et al., 2025a). A *process reward* model (PRM) is trained to score $a_h$ at $\mathbf{s}_h = (\mathbf{x}, a_1, \ldots, a_{h-1})$ either using human annotations (Lightman et al., 2024) or the value functions based on LLM-generated data (Wang et al., 2024; Luo et al., 2024; Setlur et al., 2025); indeed, PRMs estimate either the likelihood of future success or the change in the likelihood of future success before and after taking $a_h$. In addition, PRMs were also developed to improve search methods (Snell et al., 2025; Wu et al., 2025), and to identify the "first pit" in an incorrect reasoning trajectory to construct preference pairs for direct preference alignment (Hwang et al., 2024; Setlur et al., 2024).

**Reinforcement learning from AI feedback.** Reinforcement learning from human feedback (RLHF) uses human-preference-aligned reward models to evaluate response quality (Christiano et al., 2017; Ziegler et al., 2019; Stiennon et al., 2020; Ouyang et al., 2022). A key barrier to scale RLHF is the need for high-quality human labels. Previous studies (Gilardi et al., 2023; Ding et al., 2023) have shown that modern LLMs exhibit strong alignment with human judgments, suggesting that AI-generated labels can serve as a viable alternative. In this context, (Bai et al., 2022) was the first to explore RLAIF, jointly optimizing helpfulness and harmlessness using both human and AI-generated labels, and (Roit et al., 2023; Kwon et al., 2023; Lee et al., 2024) showed that LLMs can produce informative reward signals for RL post-training. Our approach can leverage AI feedback to introduce response diversity within all-negative-sample groups by assigning intermediate binary rewards to reasoning steps. Indeed, one identifies the proportion of correct steps in the reasoning trajectory and use it to compute a reward $r_i \in [0, 1)$.

## B ADDITIONAL EXPERIMENTAL RESULTS

Beyond aggregate results, we provide a targeted analysis of SGPO's impact. In line with our theoretical finding that SGPO converges faster than GRPO, empirical metrics offer supporting evidence. Prior work on RLVR entropy highlights its link to performance: Cui et al. (2025) showed that lower

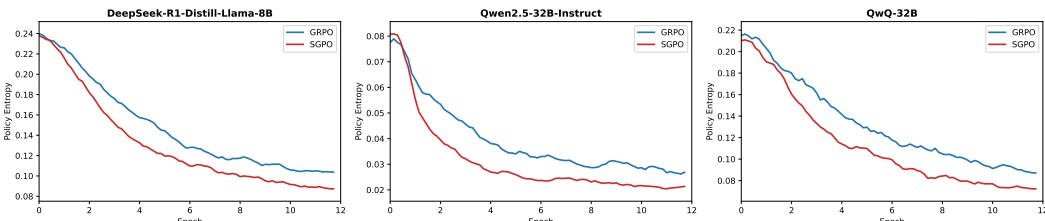

Figure 3: Policy entropy levels during training for GRPO and SGPO across different base models.

policy entropy under correct signals correlates with stronger policies, while Agarwal et al. (2025) demonstrated that directly minimizing entropy can improve performance. As shown in Figure 3, SGPO reduces policy entropy more rapidly than GRPO, indicating faster convergence toward deterministic RLVR behavior with higher rollout confidence. This matches our theoretical results, confirming that step-wise signals accelerate convergence.

## C  MISSING PROOFS

We first present the detailed setup for our stylized model and prove several technical lemmas. Then, we use these lemmas to prove our main result in Theorem 3.3.

### C.1  STYLIZED MODEL

We consider a policy parameterized by a softmax function, which is standard in the analysis of reinforcement learning methods (Agarwal et al., 2020; Mei et al., 2021; Li et al., 2024b):

$$\pi_\theta(a_{1:T} \,|\, \mathbf{x}) = \prod_{t=1}^{T} \pi_{\theta_t}(a_t \,|\, \mathbf{x}, a_{1:t-1}) = \prod_{t=1}^{T} \frac{\exp(\theta_t^{\mathbf{x}, a_{1:t-1}, a_t})}{\sum_{a_t' \in \mathcal{V}^\star} \exp(\theta_t^{\mathbf{x}, a_{1:t-1}, a_t'})},$$

By convention, we assume that $\pi_{\theta_1}(a_1 \,|\, \mathbf{x}, a_{1:0}) = \pi_{\theta_1}(a_1 \,|\, \mathbf{x})$.

For simplicity, we perform our analysis in the likelihood space rather than in the parameter space (i.e., $\theta$) directly. Indeed, we define the key quantities as follows,

$$p \doteq \pi_{\theta_1}(a_1 = 2 \,|\, \mathbf{x}) = \frac{e^{\theta_1^{\mathbf{x},2}}}{e^{\theta_1^{\mathbf{x},1}} + e^{\theta_1^{\mathbf{x},2}}}, \qquad q \doteq \pi_{\theta_2}(a_2 = 2 \,|\, \mathbf{x}, a_1 = 2) = \frac{e^{\theta_2^{\mathbf{x},2,2}}}{e^{\theta_2^{\mathbf{x},2,1}} + e^{\theta_2^{\mathbf{x},2,2}}}.$$

Note that the original 4-dimensional parameter space defined by $\theta_1^{\mathbf{x},1}, \theta_1^{\mathbf{x},2}, \theta_2^{\mathbf{x},2,1}$ and $\theta_2^{\mathbf{x},2,2}$ in $\mathbb{R}$ is reduced to a 2-dimensional likelihood space defined by $p, q \in [0, 1]$.

We rewrite the generic GRPO update with a step size $\eta > 0$ as follows,

$$\theta^{(k+1)} = \theta^{(k)} + \eta \cdot g(\theta), \quad \text{where } g(\theta) = \frac{1}{NGH} \left( \sum_{i=1}^{N} \sum_{k=1}^{G} \sum_{h=1}^{H} s_\theta(\mathbf{x}^i, a_{1:h-1}^{i,k}) A_{i,k} \right),$$

where $N$ is the number of prompts, $G$ is the number of groups, $H$ is the number of reasoning steps in each response, $s_\theta(\mathbf{x}^i, a_{1:h-1}^{i,k}) := \nabla_\theta \log \pi_\theta(a_t|\mathbf{x}, a_{1:h-1})$ is the score function, and the advantage $A_{i,k}$ is defined by

$$A_{i,k} = \frac{r(\mathbf{x}^i, \mathbf{y}^{i,k}) - (1/G) \sum_{j=1}^{G} r(\mathbf{x}^i, \mathbf{y}^{i,j})}{\sqrt{(1/G) \sum_{j=1}^{G} (r(\mathbf{x}^i, \mathbf{y}^{i,j}) - (1/G) \sum_{j'=1}^{G} r(\mathbf{x}^i, \mathbf{y}^{i,j'}))^2}}$$

To distinguish, we denote $g_{\text{GRPO}}(\cdot)$ as the gradient estimator using classical outcome reward model $r$, and $g_{\text{SGPO}}(\cdot)$ as the gradient estimator using the reward $r_{\text{SGPO}}$ as proposed in Section 3.1.

For our simple stylized model, we compute the score functions in terms of likelihood parameters $p, q$ as follows,

$$s(a_1 = 1 \,|\, \mathbf{x}) = \begin{bmatrix} p \\ -p \\ 0 \\ 0 \end{bmatrix}, \quad s(a_1 = 2 \,|\, \mathbf{x}) = \begin{bmatrix} p - 1 \\ 1 - p \\ 0 \\ 0 \end{bmatrix},$$

and

$$s(a_2 = 1 \mid \mathbf{x}, a_1 = 2) = \begin{bmatrix} 0 \\ 0 \\ q \\ -q \end{bmatrix}, \quad s(a_2 = 2 \mid \mathbf{x}, a_1 = 2) = \begin{bmatrix} 0 \\ 0 \\ q-1 \\ 1-q \end{bmatrix}.$$

Note that we restrict the sample space to $\mathbf{y} \in \{(1,1), (2,1), (2,2)\}$, excluding the sample $(1, 2)$. The responses can be drawn i.i.d. from the distribution as follows,

$$(a_1, a_2) = \begin{cases} (1,1), & \text{w.p. } 1-p, \\ (2,1), & \text{w.p. } p(1-q), \\ (2,2), & \text{w.p. } pq. \end{cases}$$

We set $G = 2$ and focus on the SGPO and GRPO training dynamics with population-level policy gradient which can be computed exactly for the stylized model as follows,

$$\bar{g}_{\text{SGPO}}(\theta) = \mathbb{E}[g_{\text{SGPO}}(\theta)] = \tfrac{1}{2} \begin{bmatrix} p(p-1) \\ p(1-p) \\ p^2 q(q-1) \\ p^2 q(1-q) \end{bmatrix}, \quad \bar{g}_{\text{GRPO}}(\theta) = \mathbb{E}[g_{\text{GRPO}}(\theta)] = \tfrac{1}{2} \begin{bmatrix} p(p-1)q \\ p(1-p)q \\ pq(q-1) \\ pq(1-q) \end{bmatrix}.$$

Since $g_{\text{GRPO}}(\theta)$ and $g_{\text{SGPO}}(\theta)$ concentrate around $\bar{g}_{\text{GRPO}}(\theta)$ and $\bar{g}_{\text{SGPO}}(\theta)$ when the number of samples in each group is sufficiently large, it is reasonable to analyze the population-level dynamics at first. Note that the high-probability guarantees for the sample-level dynamics can be derived using concentration inequalities under certain conditions.

We can explicitly write down the SGPO and GRPO update rules with $\eta = 1$ using the likelihood parameters $p$ and $q$ as follows,

$$\begin{cases} p_{\text{SGPO}}^{(k+1)} = \exp(f_{11}(p_{\text{SGPO}}^{(k)})), \\ q_{\text{SGPO}}^{(k+1)} = \exp(f_{12}(p_{\text{SGPO}}^{(k)}, q_{\text{SGPO}}^{(k)})), \end{cases} \quad \text{and} \quad \begin{cases} p_{\text{GRPO}}^{(k+1)} = \exp(f_{21}(p_{\text{GRPO}}^{(k)}, q_{\text{GRPO}}^{(k)})), \\ q_{\text{GRPO}}^{(k+1)} = \exp(f_{22}(p_{\text{GRPO}}^{(k)}, q_{\text{GRPO}}^{(k)})), \end{cases} \quad (3)$$

where the functions $f_{ij}$ are defined by

$$\begin{aligned} f_{11}(p) &= \log(p) + p(1-p) - \log(1 - p + pe^{p(1-p)}), \\ f_{21}(p, q) &= \log(p) + p(1-p)q - \log(1 - p + pe^{p(1-p)q}), \\ f_{12}(p, q) &= \log(q) + p^2 q(1-q) - \log(1 - q + qe^{p^2 q(1-q)}), \\ f_{22}(p, q) &= \log(q) + pq(1-q) - \log(1 - q + qe^{pq(1-q)}). \end{aligned} \quad (4)$$

## C.2 TECHNICAL LEMMAS

We provide several technical lemmas that are important to the subsequent proof of Theorem 3.3. Indeed, the first lemma summarizes the properties of particular functions related to the aforementioned functions $f_{11}$, $f_{21}$, $f_{12}$ and $f_{22}$ from Eq. (4).

**Lemma C.1.** *The following statements hold true,*

*(i) The function $f_{11}$ is strictly increasing on $(0, 1)$.*
*(ii) The function $h_p(x) := x - \log(1 - p + pe^x)$ is strictly increasing for any fixed $p \in (0, 1)$.*
*(iii) The function $f_{21}$ is strictly increasing in either $p$ for any fixed $q$ or $q$ for any fixed $p$ on $(0, 1)$.*
*(iv) The function $\varphi(x) := \log(1 + (1/2)e^{-e^x})$ is strictly concave on $(-\infty, 0)$.*

*Proof.* First of all, we have

$$f_{11}'(p_1) = \frac{1 + (1 - 2p_1)p_1(1 - p_1)}{p_1(1 + p_1 + p_1 e^{p_1(1-p_1)})} > \frac{3}{4p_1(1 + p_1 + p_1 e^{p_1(1-p_1)})} > 0.$$

Thus, the function $f_{11}$ is strictly increasing on $(0, 1)$.

Furthermore, we have

$$h_p'(x) = 1 - \frac{pe^x}{1 - p + pe^x} = \frac{1-p}{1-p+pe^x} \overset{0<p<1}{>} 0.$$

Thus, the function $h_p(x)$ is strictly increasing.

Moreover, we have

$$\frac{\partial f_{21}(p_1,p_2)}{p_1} = \frac{1+p_2(1-2p_1)p_1(1-p_1)}{p_1(1+p_1+p_1e^{p_2p_1(1-p_1)})} > \frac{3}{4p_1(1+p_1+p_1e^{p_1(1-p_1)})} > 0,$$

$$\frac{\partial f_{21}(p_1,p_2)}{p_2} = \frac{p_1(1-p_1)^2}{1+p_1+p_1e^{p_2p_1(1-p_1)}} > 0.$$

Thus, the function $f_{21}$ is strictly increasing in either $p$ for any fixed $q$ or $q$ for any fixed $p$ on $(0,1)$.

Finally, we have

$$\varphi''(x) = \frac{(e^{x+e^x}/2 - e^{e^x}/2 - 1/4)e^x}{e^{2e^x} + e^{e^x} + 1/4}.$$

Since $u = e^x \in (0,1)$ for $x < 0$, we have

$$(ue^u/2 - e^u/2 - 1/4)u = (e^u(u-1)/2 - 1/4)u < -(1/4)u < 0.$$

Thus, $\varphi''(x) < 0$ for all $x < 0$ which shows that $f$ is strictly concave on $(-\infty, 0)$. $\square$

The second lemma presents an inequality which plays a key role in the proof of Theorem 3.3.

**Lemma C.2.** *We define the auxiliary functions as follows,*

$$A(x) = 1 + \left(\frac{1}{x} - 1\right)e^{-x(1-x)}, \ B(x,y) = 1 + \left(\frac{1}{y} - 1\right)e^{-x^2y(1-y)},$$

$$C(z) = 1 + \left(\frac{1}{z} - 1\right)e^{-z^2(1-z)}.$$

*Then, we have $C(\sqrt{xy})^2 > A(x)B(x,y)$ for all $x$ and $y$ satisfying $1/2 < y < x < 1$.*

*Proof.* We consider the lower and upper bound of $e^{-u}$ when $u > 0$:

$$1 - u + \frac{u^2}{2} - \frac{u^3}{6} < e^{-u} < 1 - u + \frac{u^2}{2}.$$

Since $1/x - 1$, $1/y - 1$ and $1/\sqrt{xy} - 1$ are all positive, we have

$$A(x) \le 1 + \frac{1-x}{x}\left(1 - x(1-x) + \frac{x^2(1-x)^2}{2}\right) = \frac{1}{x} - (1-x)^2 + \frac{x(1-x)^3}{2}.$$

$$B(x,y) \le 1 + \frac{1-y}{y}\left(1 - x^2y(1-y) + \frac{x^4y^2(1-y)^2}{2}\right) = \frac{1}{y} - x^2(1-y)^2 + \frac{x^4y(1-y)^3}{2}.$$

$$C(z) \ge 1 + \frac{1-z}{z}\left(1 - z^2(1-z) + \frac{z^4(1-z)^2}{2} - \frac{z^6(1-z)^3}{6}\right)$$

$$= \frac{1}{z} - z(1-z)^2 + \frac{z^3(1-z)^3}{2} - \frac{z^5(1-z)^4}{6}.$$

Set $z^2 = xy$, the original statement is equivalent to $(zC(z))^2 > (xA(x))(yB(x,y))$. Using the above upper and lower bound, it suffices to show $C_1(\sqrt{xy})^2 > A_1(x)B_1(x,y)$ where

$$A_1(x) = 1 - x(1-x)^2 + x^2(1-x)^3/2, \ B_1(x,y) = 1 - x^2y(1-y)^2 + x^4y^2(1-y)^3/2,$$

$$C_1(z) = 1 - z^2(1-z)^2 + z^4(1-z)^3/2 - z^6(1-z)^4/6.$$

By Lemma C.3, this is indeed true. This completes the proof. $\square$

**Lemma C.3.** *Define functions*

$$A_1(x) = 1 - x(1-x)^2 + x^2(1-x)^3/2, \ B_1(x,y) = 1 - x^2y(1-y)^2 + x^4y^2(1-y)^3/2,$$

$$C_1(z) = 1 - z^2(1-z)^2 + z^4(1-z)^3/2 - z^6(1-z)^4/6.$$

*Then, $C_1(\sqrt{xy})^2 > A_1(x)B_1(x,y)$ for all $1/2 < y < x < 1$.*

*Proof.* Let $x = u^2$ and $y = v^2$, then $1 > u > v > 1/\sqrt{2}$ and $z = uv$. We next show the desired inequality holds on a larger region, i.e., $1 > u > v > 2/3$. On this larger region, we have the reparameterization as follows:

$$u = \frac{2s+3}{3s+3}, \quad v = \frac{2r+2s+3}{3r+3s+3}, \quad s, r \in (0, +\infty),$$

or equivalently,

$$s = \frac{3(1-u)}{3u-2}, \quad r = \frac{3(u-v)}{(3u-2)(3v-2)}, \quad 1 > u > v > \frac{2}{3}.$$

It is easy to see this defines a one-to-one correspondence from $(u, v)$-space to $(s, r)$-space. Thus, we aim to prove the following function $f$ is positive:

$$F(s,r) := C_1 \left( \frac{2s+3}{3s+3} \cdot \frac{2r+2s+3}{3r+3s+3} \right)^2 - A_1 \left( \left( \frac{2s+3}{3s+3} \right)^2 \right) B_1 \left( \left( \frac{2s+3}{3s+3} \right)^2, \left( \frac{2r+2s+3}{3r+3s+3} \right)^2 \right).$$

By leveraging Sympy's symbolic engine, the function expands and simplifies to:

$$F(s,r) = \frac{f(s,r)}{c(s+1)^{20}(r+s+1)^{20}}, \text{ where } f(s,r) := \sum_{k=0}^{20} c_{20-k}(s) r^{20-k},$$

where $c > 0$ is a universal constant, and single-variable polynomials $c_{20}(s), \ldots, c_2(s), c_0(s) > 0$ and $\Delta_2 := c_1(s)^2 - 4c_2(s)c_0(s) < 0$, for all $s > 0$ (see Table 6 for details). Notice that from the table, we can see the only nontrivial parts are $c_3(s) > 0$ and $c_2(s) > 0$ because only these two contain negative coefficients. The positivity of $c_3(s)$ is simple because there is only one term ($s^9$) with negative coefficient and for all $s > 0$,

$$19471456710454363005152664s^{10} + 9684588377731643071927236s^8 > 14413823109350224541499726s^9.$$

To see this, simple estimation and AM-GM inequality yield

$$\text{LHS} > 1.9 \times 10^{25} s^{10} + 9.6 \times 10^{24} s^8 > 2\sqrt{182.4} \times 10^{24} s^9 > 2.7 \times 10^{25} s^9 > \text{RHS}.$$

The positivity of $c_2(s)$ is more complicated because it has 4 negative terms $s^{10}, s^9, s^8, s^7$. However, we can use similar idea, i.e., choosing a pair of positive terms to bound a negative term:

$$95791062786555508724088742320s^{15} + 571809550541807937530952s^5 > 7027359523643236832970716s^{10};$$

$$43785862330162499052209529768s^{14} + 18478934378953446115 0530s^4 > 9985407270432287153739 2604s^9;$$

$$16326736853527122991715155824s^{13} + 35488375569622472169240s^3 > 3572603137796979208818 8925s^8;$$

$$46082190500843267907489 33153s^{12} + 4362950858813170449228s^2 > 60990378953076701422876 08s^7.$$

To see this, simple estimation and AM-GM inequality yield

$$\text{LHS} > 9.5 \times 10^{28} s^{15} + 5.7 \times 10^{23} s^5 > 2\sqrt{541.5} \times 10^{25} s^{10} > 4.6 \times 10^{26} s^{10} > \text{RHS};$$

$$\text{LHS} > 4.3 \times 10^{28} s^{14} + 1.8 \times 10^{23} s^4 > 2\sqrt{77.4} \times 10^{25} s^9 > 1.7 \times 10^{26} s^{10} > \text{RHS};$$

$$\text{LHS} > 1.6 \times 10^{28} s^{13} + 3.5 \times 10^{22} s^3 > 2\sqrt{5.6} \times 10^{25} s^9 > 4.7 \times 10^{25} s^8 > \text{RHS};$$

$$\text{LHS} > 4.6 \times 10^{27} s^{12} + 4.3 \times 10^{21} s^2 > 2\sqrt{19.7} \times 10^{24} s^7 > 8.8 \times 10^{24} s^7 > \text{RHS}.$$

In conclusion, we have all coefficient $c_i(s)$ positive except $c_1(s)$, but it doesn't affect the positivity of $f$ because $\Delta_2 < 0$. This completes the proof. $\square$

Table 6: Coefficient Lists of $F(s, r)$

| Notation | Value |
|---|---|
| $c$ | 4376759565260494436836 |
| $c_{20}$ | $2(2s + 3)^2(171477432074484750s^{18} + 266104092606915762200s^{17} + 191778746468802317181s^{16} + 850194149855082319224s^{15} + 258708243429004580649s^{14} + 5704415906039160731874s^{13} + 9366197581963232054460s^{12} + 11563054951307567026248s^{11} + 10670965452123886149660s^{10} + 7187176769582075261292s^9 + 3372972168996579430017s^8 + 1072082836158220703952s^7 + 370302094042890771285s^6 + 329901677376902425818s^5 + 2827999868056166267862s^4 + 15116827017089336365008s^3 + 49139849518345513368s^2 + 9090603727935062976s + 742484948385838248)$ |
| $c_{19}$ | $8(2s + 3)^2(8573871603724243750s^{19} + 14118375184879884 0450s^{18} + 1085065457535611084097s^{17} + 5159905424662678527663s^{16} + 16962017821014041355285s^{15} + 40761515892906930261393s^{14} + 73777358861762677983126s^{13} + 101968163476291942277643s^{12} + 107719550183279202945336s^{11} + 85951871005332247942347s^{10} + 50477142420763872747039s^9 + 21228461710484227270812s^8 + 7175576253360286202193s^7 + 3821642119447908810138s^6 + 33431406025396150 70982s^5 + 2308058741380310946144s^4 + 10465286915656214 71344s^3 + 300769285860744146028s^2 + 50403014643440592936s + 3785769852305984190)$ |
| $c_{18}$ | $2(2s + 3)^2(325807120941521262500s^{20} + 5673987380977312396200s^{19} + 4632014361420018357 5358s^{18} + 235164624868455434314740s^{17} + 830330782346499878631402s^{16} + 2159105892766696625508432s^{15} + 4268066364777710628878112s^{14} + 6521177726276586191628264s^{13} + 7742933419720025359660131s^{12} + 7110911361873582109051992s^{11} + 4976474876383070663195517s^{10} + 2600230719135591148269222s^9 + 1035307928116150386109695s^8 + 420495220158165783672300s^7 + 2874363458621680 26209421s^6 + 225749912117527047120354s^5 + 132205553924765757023286s^4 + 52546262098747895532864s^3 + 13596497258861930544108s^2 + 2087305777245729888936s + 145293329180967197454)$ |
| $c_{17}$ | $12(2s + 3)^2(325807120941521262500s^{21} + 5982992191700268855300s^{20} + 51700930512613327403406s^{19} + 279079780777259477944590s^{18} + 1053187317945045364767966s^{17} + 2945458902809849586876810s^{16} + 6310900331865363012743844s^{15} + 10554182290179327214273482s^{14} + 13894164004300292404029789s^{13} + 14397511755502695216399777s^{12} + 11648187532971253859583195s^{11} + 7253672843249894875203621s^{10} + 3463934076062711984148183s^9 + 1401311871124636922510679s^8 + 709635073453803384330663s^7 + 526601300781722718969621s^6 + 366411462920903557014120s^5 + 187790938366917450633606s^4 + 66828265788979238418684s^3 + 15774993964318985462592s^2 + 2238177282159012945966s + 145311275743961970078)$ |
| $c_{16}$ | $3(2s + 3)^2(55387210560058614625 00s^{22} + 1069639490411948303 44800s^{21} + 975372417387075095557110s^{20} + 5577701869567635624516312s^{19} + 22401070138854745562671602s^{18} + 67034725561809321462257220s^{17} + 154689592660061775780683034s^{16} + 280889539801332767094091608s^{15} + 405664328993005936098158220s^{14} + 467432234597450323377654624s^{13} + 428162911701836470453488816s^{12} + 308851193177419859648120664s^{11} + 173923711507624595412555792s^{10} + 78611043902295277305014364s^9 + 345027239321086599680 68191s^8 + 207848696780878470113505 12s^7 + 152110704912752708992604 79s^6 + 94397951694788177205573 90s^5 + 43236427772992108674668 40s^4 + 1398336255225225668686176s^3 + 30416510338341914536668 0s^2 + 40170170984468888396880s + 2445688799534592091926)$ |

| | |
|---|---|
| $c_{15}$ | $12(2s + 3)^2(4430976844804689170000s^{23} + 89773624658788072119600s^{22} + 861449789967478967902968s^{21} + 5202052340570363961799272s^{20} + 22150958674722147636887880s^{19} + 70610768977431597138502416s^{18} + 174547747719038741579249148s^{17} + 341846798830220759744006802s^{16} + 537017897494050473220887475s^{15} + 680386317088643909072652357s^{14} + 694900917445527709102606203s^{13} + 568854294351452352559155357s^{12} + 370162500862325208186949407s^{11} + 192089934615274810143113637s^{10} + 85560000374404871078044743s^9 + 42188979069325595690484894s^8 + 27986482983635448916149477s^7 + 19349473000527948423160098s^6 + 10826549522417650582347903s^5 + 4499119009419911850147903s^4 + 1337268081929646109116429s^3 + 270188727447397870150299s^2 + 33412191448722202871793s + 1916481339467789227047)$ |
| $c_{14}$ | $3(2s + 3)^2(44309768448046891700000s^{24} + 939760900846202799633600s^{23} + 9465882231449979270581616s^{22} + 60189415644287265430240224s^{21} + 270831851136366961169859120s^{20} + 916082243422713193340650464s^{19} + 2414605718335618880853970032s^{18} + 5071784805050410035823167576s^{17} + 8606003351786628019139811024s^{16} + 11882072721559268002578594336s^{15} + 13373135736066588915267225192s^{14} + 12233789753017327381398656592s^{13} + 9038087243602138768195078704s^{12} + 5369286552690873446276117328s^{11} + 2623535852573549267091555300s^{10} + 1196071780982939651865144096s^9 + 660078244623496780263588075s^8 + 451393195997666208667458852s^7 + 290343677073268941864046305s^6 + 148042042832629803967321050s^5 + 56464193331043404116741784s^4 + 15559478430192850829818824s^3 + 2939179015494775847121192s^2 + 342046929585497061253176s + 18557914646800278459054)$ |
| $c_{13}$ | $6(2s + 3)^2(44309768448046891700000s^{25} + 981785555104524878071200s^{24} + 10357132334422169539112688s^{23} + 69166131783384274997062320s^{22} + 327906609514495942625889552s^{21} + 1172872936442019296825004672s^{20} + 3283063721629310545911589320s^{19} + 7360333239220785966714322212s^{18} + 13411238091959519801173855314s^{17} + 20030874331258639447616405430s^{16} + 24612632066584435063045693896s^{15} + 24861807943243247848564702728s^{14} + 20554616309938676564088193632s^{13} + 13828701774644445607198489296s^{12} + 7593307018019776947591316692s^{11} + 3573164437682539651140298914s^{10} + 1711205400898634741432588709s^9 + 1026436201325482873227731181s^8 + 693346698158130213351442587s^7 + 413729468327452341092783823s^6 + 194077643830831926335960674s^5 + 68561905220231775619051080s^4 + 17640970508803332546397944s^3 + 3132633769453198732801032s^2 + 344560424227000935565860s + 17745096925489168423620)$ |
| $c_{12}$ | $3(2s + 3)^2(144006747456152398025000s^{26} + 3327383180429252608653600s^{25} + 368686806677921921575024412s^{24} + 256712663785782687868607040s^{23} + 1278878488443918869718977532s^{22} + 4822501965635852078399708760s^{21} + 14284937167605407625123613032s^{20} + 34039675734020531218713639960s^{19} + 66269194087347804842641890936s^{18} + 106420251181999059483739591464s^{17} + 141673718347886548611527896944s^{16} + 156512253734744849831503592064s^{15} + 143119458698672790284020359156s^{14} + 107772919457957006535562903236s^{13} + 66583563316818931905117287625s^{12} + 3420802797787101480006575072s^{11} + 15821430510294339891416781741s^{10} + 8030913522092664743057080482s^9 + 5050521785734143734257145460s^8 + 3292374301511579644939102872s^7 + 1827212365211212115171329068s^6 + 795146625577647417589837740s^5 + 262142182103903879108812389s^4 + 63354427214296180937779584s^3 + 10625610419721898094320272s^2 + 1108770079900593722922360s + 543716225994186505 21707)$ |

| | |
|---|---|
| $c_{11}$ | $2(2s + 3)^2(2880134949123047960500000s^{27} + 69279266135375987271516000s^{26} + 79684994050724088696229560s^{25} + 58301357534867002173271898^{24} + 30447196677965557178181595444s^{23} + 1207115288169085268602840692000s^{22} + 377197518022830645618602283122^{21} + 951865707546849789543156807244s^{20} + 1971414432505091371211002980188s^{19} + 3386213982946173069363008884866s^{18} + 4853203941721254940783622901422s^{17} + 5817879064516111538357630359266s^{16} + 5828027860833244408675746802192s^{15} + 4859938328494178086778700964800s^{14} + 3355944446899925164463735414700s^{13} + 1918186782349513132753181871666s^{12} + 93215508141687941579846043591s^{11} + 4299548855661907296542112784500s^{10} + 2299333342054457044986385516590s^9 + 14702002876383491619167125293s^8 + 9151802478157350485470780638s^7 + 47464951715999981131404092944s^6 + 19302388834938005528214678366s^5 + 597661705826550277052178582s^4 + 1363694210951181524092878750s^3 + 2169005140767851682417301500s^2 + 21544004474439075954871833s + 10087313175869404602874500)$ |
| $c_{10}$ | $(2s + 3)^2(63362968880707055131000000s^{28} + 158423911056767229213912000s^{27} + 189762104968446645014104296s^{26} + 14488992334199058656962307044s^{25} + 79150177490376318122969892722s^{24} + 3291137409296388117112885180800s^{23} + 10818411368041983695706257112000s^{22} + 28818145740591320391556014765600s^{21} + 63256307754712576828972417585200s^{20} + 115696624884216722807785375867200s^{19} + 177561023038915209184039832629200s^{18} + 229466176991789299757557690348800s^{17} + 24981929658677011622686888359244s^{16} + 22856591123054892544345187114160s^{15} + 1748991971403770816761602941934s^{14} + 11137976841637858426496732337600s^{13} + 5924913951597435743900967421110s^{12} + 2744766007155415656150112137960s^{11} + 1270446195852713506501888454520s^{10} + 70538582326147723865407765680s^9 + 44889681102540017067072498465s^8 + 26602898887160759567836334520s^7 + 12967769746561749479001701280s^6 + 4960789745148078555411299844s^5 + 14507320987315900495954230840s^4 + 3139308944683290821452849560s^3 + 47525670620612611058618019s^2 + 4506795195444098216749962s + 2019810664389200880741210)$ |
| $c_9$ | $2(2s + 3)^2(28801349491230479605000000s^{29} + 74742471188957857468404000s^{28} + 93085137978826092989684184s^{27} + 74040224418389061361023861600s^{26} + 42224920164363414473174228400s^{25} + 18373474587737757143576597976s^{24} + 6337431126498352669577742938400s^{23} + 17768929619898477040864660549200s^{22} + 41199212984768386445385597789000s^{21} + 799270570087512158479167778014s^{20} + 13074545632659705306616623104880s^{19} + 18114409066735078501962269572920s^{18} + 21290088336315518034044019666180s^{17} + 21203189029974651214705528123980s^{16} + 17827525781427740408821260458460s^{15} + 12583486367559299555587480574282s^{14} + 7422229506057785006481351208080s^{13} + 3687595326777177603238716204480s^{12} + 1632923401776166653452978780340s^{11} + 75628135544208165254280371040s^{10} + 4285873595461389310248944256900s^9 + 268156098604935554390171069670s^8 + 15154066307595953401565854680s^7 + 6986502898234763627448498006s^6 + 25297749386947476621523773630s^5 + 70233050181067477048786272300s^4 + 14473265502786394882786226100s^3 + 2092454720837126248556029500s^2 + 189952977760871511452139900s + 8166786538840395351817500)$ |

| | |
|---|---|
| $c_8$ | $3(2s + 3)^2(144006747456152398025000s^{30} + 387370368578743962834 2400s^{29} + 500869506060239258684 79036s^{28} + 414347293343103311413357200s^{27} + 246245571641789555346681 4404s^{26} + 111903118390663442595367 14024s^{25} + 4040929597277317705010456 6556s^{24} + 118947134811306669490272616200s^{23} + 29046044800144993798660964 1984s^{22} + 595650982876961700444247916136s^{21} + 103439450333283704483287197 3660s^{20}+152912252239834145223774826 4328s^{19}+192917011214440853129524724 0352s^{18}+207727572983577989862797245 2564s^{17}+190440009258484914739880325 7857s^{16}+147929501924676524442161796 0408s^{15}+96723258758152313412876769 0737s^{14} + 52982714945629451357986685 7762s^{13} + 245850149371151220247602350838s^{12} + 1035861819814159623650793501 36s^{11} + 47441969831005210765856651 490s^{10} + 27017370328613731242988726 116s^{9} + 16543604074621342316534833896s^{8} + 8959808104815503968120901160s^{7} + 39341484805034175878771776 53s^{6} + 1356708501926603220744080694s^{5} + 3593560100508123307874732 79s^{4} + 70800398025365428901919252s^{3} + 98057056394418600109930 74s^{2} + 85429442798900619720505 2s + 3530618607561154024305 6)$ |
| $c_7$ | $12(2s + 3)^2(221548842240234458500 00s^{31} + 616966740327228674348400s^{30} + 82709070736961626834 30488s^{29} + 710548883749584474193 31400s^{28} + 439319157110546803811896968s^{27} + 208115303570456665813062 4800s^{26} + 785169940845868080224 2634484s^{25} + 242075167606254230722 07073414s^{24} + 620925174288058758667946 74881s^{23} + 134191349159021239688414857431s^{22} + 246518378732311775029464265197s^{21} + 387229048800653531738114739999s^{20} + 521849077561894657636398449655s^{19} + 604010499659026951460897607741s^{18} + 599706193952136964321127694249s^{17} + 508932600092700362308014694866s^{16} + 366923845830106735087199088303s^{15} + 222970678878271885535159028828s^{14} + 113510751404816691589761355815s^{13} + 489596215801597338452898846 33s^{12} + 193438197331709555443195521 63s^{11} + 857096764824341929866037721 1s^{10} + 484228170521352710841651327 9s^{9} + 29091383745623179903600313 94s^{8} + 15229213035961739961385759 05s^{7} + 642212965528640572310436 879s^{6} + 212347408792609836644815 359s^{5} + 539405048691381659833555 56s^{4} + 102009814215488662722720 21s^{3} + 1357609423748226427157778s^{2} + 113782587779245747678886 5s + 4528476210896135134182)$ |
| $c_6$ | $3(2s + 3)^2(44309768448046891700000s^{32} + 1275958134912779427134400s^{31} + 1771212464874352037424 6000s^{30} + 157800783303192342484807776s^{29} + 1013476002096822653934247536s^{28} + 499634681147199443824097 0400s^{27} + 19656891049003759997738219664s^{26} + 63343018585986823082619729288s^{25} + 170258537043632779229473723584s^{24} + 386719255970188993917224574336s^{23} + 749195973037122202127202135768s^{22}+1245952789865163842159925500256s^{21}+1785990969360715046000117363568s^{20}+2210911897085152430737822315488s^{19}+2363325656949833631869930474232s^{18}+2176386534475868543250178452024s^{17}+1718519269597730769046219931925s^{16}+1154898997656814617038309424924s^{15}+653993799682370716585040976045s^{14} + 309097812387589842994049088882s^{13} + 1226499865148059279758110136 78s^{12} + 44139321593386486437541304232s^{11} + 1819460051207313117955247421 8s^{10} + 10080729455247177149466360108s^{9} + 6012154495009832863143860703s^{8} + 3086713901904485302077423912s^{7} + 1264536513002644891553573532s^{6} + 404269004313535997709064188s^{5} + 99085318554137395893467265s^{4} + 18066869592965845573731768s^{3} + 23180375592196844545338 93s^{2} + 187335877784959577298978s + 7192069815364796066229)$ |

| | |
|---|---|
| $c_5$ | $6(2s + 3)^2(886195368960937834000s^{33} + 263596557834220301114400s^{32} + 3784460184258343211722032s^{31} + 34920466929143934879808368s^{30} + 232642130166877120425752976s^{29} + 1191697800945367086458055072s^{28} + 4880694751880441559218476632s^{27} + 16406950824469547680560778764s^{26} + 46112865280418851489322074134s^{25} + 109812274944960351155991786018s^{24} + 223725856433516713998849805632s^{23} + 392660140917098227883941019880s^{22} + 596450332899276085006754000436s^{21} + 786241559139425306167747437324s^{20} + 900162472445706270827205735408s^{19} + 894118456198813762374744874842s^{18} + 768019765700356821541048064211s^{17} + 567188919515029404795124639815s^{16} + 356892849173690588784279865923s^{15} + 188887726440058090918123569807s^{14} + 82758519677573937797801658678s^{13} + 29755841968295155180938377160s^{12} + 9265655123985315388229427762s^{11} + 3269804705124270960946179492s^{10} + 1745378750437674672254525343s^{9} + 1067124171571276165079329161s^{8} + 553192327447935426146669544s^{7} + 224471881547232310097558892s^{6} + 70271287113829333092849234s^{5} + 16759856517295120894936656s^{4} + 2963427571961631174417201s^{3} + 367988093962382467875321s^{2} + 28750493973686584294998s + 10663388997876680421860)$ |
| $c_4$ | $3(2s + 3)^2(5538721056005861462500s^{34} + 170000930428677948001200s^{33} + 2521542870629614052494182s^{32} + 24069033861073900393194288s^{31} + 166112140239471730224127950s^{30} + 882861984743248195220227068s^{29} + 3758135559481861075860190560s^{28} + 13155787658258741235890755800s^{27} + 38587134973448786366330248032s^{26} + 96129109345583765886638321232s^{25} + 205447147746380335664768754954s^{24} + 379449453375926778871028692368s^{23} + 608776780980435381145177931646s^{22} + 851251839731897497067600541072s^{21} + 1039128387112340409627618422211s^{20} + 1107338907683241287910817574616s^{19} + 1028236425910911889110211542327s^{18} + 828653388719550997841334599814s^{17} + 575679860099214460991336767122s^{16} + 341113980250874385715195498752s^{15} + 169602112160966873244672554562s^{14} + 69019121880524178108395152092s^{13} + 22192275965509681657763172138s^{12} + 5513039324469607430622101184s^{11} + 1308552733582056616685717799s^{10} + 622822426968272935610750562s^{9} + 441208218615564366021478680s^{8} + 251086626744997918806221832s^{7} + 105542726343197962447668330s^{6} + 33182712041751038192684904s^{5} + 7819078036046387444720541s^{4} + 1353372939412637489000388s^{3} + 163593372233172281873202s^{2} + 12396727299661082512092s + 444813790293506728368)$ |
| $c_3$ | $6(2s + 3)^2(651614241883042525000s^{35} + 206181119083643318565400s^{34} + 315621334999692786151116s^{33} + 3113065178572047666007860s^{32} + 22229797545183952680520284s^{31} + 122422561739941481599835484s^{30} + 540840514528742111761776912s^{29} + 1968384017248063183323104976s^{28} + 6014334916396782599305697730s^{27} + 15642733509059107994947830402s^{26} + 34991063427920339849664070098s^{25} + 67834512242456744868412053510s^{24} + 114610735641635823232680558264s^{23} + 169420289219412443682245006436s^{22} + 219630129457905943676457475806s^{21} + 249914888458917575863280485878s^{20} + 249458032509437740525240777731s^{19} + 217920380868106493481515756421s^{18} + 165868299839291545410161133771s^{17} + 109196804131621756456248426789s^{16} + 61455743971044438451107683232s^{15} + 29010820857450160933812140136s^{14} + 11111724769820435809112639700s^{13} + 3230978841571830756780993930s^{12} + 597734071316630030096071452s^{11} + 19471456710454363005152664s^{10} - 14413823109350224541499726s^{9} + 9684588377731643071927236s^{8} + 12560219583039185504039910s^{7} + 6629813221106696409536742s^{6} + 2267528918316638233964400s^{5} + 549510851612447197946100s^{4} + 95169593716835317546698s^{3} + 11333379076369208961606s^{2} + 837761545501050427146s + 29118359247617829474)$ |

| | |
|---|---|
| $c_2$ | $(2s + 3)^2(651614241883042525000s^{36} + 2123612870508923148 3600s^{35} + 335176828913504517258492s^{34} + 341248027234906031459 0760s^{33} + 2518418762184767430421861 2s^{32} + 1435324779126332041071110 40s^{31} + 65719953341348773774600584 0s^{30} + 2483057802762141981372604872s^{29} + 7890444256414221204984097182s^{28} + 21386706626264438997499257504s^{27} + 49968626609865030 54943532722s^{26} + 10144394420753412495 2038283748s^{25} + 180022708498134425424774257358s^{24} + 28047331654865333 0703764939232s^{23} + 384770188717561290211106882724s^{22} + 46556806140902534614 0795055268s^{21} + 497070528817890647576496874221s^{20} + 467853455550722338016560291068s^{19} + 38725509163924536134275 2551670s^{18} + 280658492377792155024177 61676s^{17} + 176840580743027422215976620477s^{16} + 957910627865555087240 88742320s^{15} + 437858623301624990522095 29768s^{14} + 163267368535271229917 15155824s^{13} + 46082190500843267907489 33153s^{12} + 7630908802856105790078 63108s^{11} - 702735952364323683297077 16s^{10} - 99854072704322871537 392604s^9 - 35726031377969792088188925s^8 - 60990378953076701422 87608s^7 + 3859970007112238323 56168s^6 + 5718095505418079375 30952s^5 + 18478943437895344611 50530s^4 + 354883755696224721 69240s^3 + 436295085881317044 9228s^2 + 32060214099439012 2456s + 1080681374138393 6712)$ |
| $c_1$ | $2s^2(2s + 3)^2(5s + 6)^2(1371819456595879000s^{33} + 4271634555705596680 88s^{32} + 643544475178313701908s^{31} + 6247566555702580389060s^{30} + 4391676555652 69249016660s^{29} + 23812943705249345217 0540s^{28} + 10360742264421251834191 68s^{27} + 371493010638125678667 1824s^{26} + 11187728687072575053763704s^{25} + 28696708254175557235886484s^{24} + 63352891406570842442956878s^{23} + 121330161626918848657558398s^{22} + 202765255131175811905486752s^{21} + 296952156159684842999476188s^{20} + 382193906007924533060066196s^{19} + 432977106373155008061033636s^{18} + 431882942555537334945182376s^{17} + 378922778321739951076319160s^{16} + 291693785851369028780662644s^{15} + 196147224158011892839433394s^{14} + 114410349624644626423212729s^{13} + 57248884332430976396451627s^{12} + 24130617950503756984051008s^{11} + 8287189086050003227856022s^{10} + 2152391916458370195915867s^9 + 325184614597411140314601s^8 - 33007972351878903475404s^7 - 41792510938686638897304s^6 - 15831185972996449730358s^5 - 3877006197061115690130s^4 - 665506024092175855680s^3 - 78419814392209911120s^2 - 5759186746951521828s - 200126180395998828)$ |
| $c_0$ | $s^4(2s + 3)^2(5s + 6)^4(5487277826383516s^{30} + 162900207047936448s^{29} + 2337377142714373098s^{28} + 21588165729897598296s^{27} + 144210704373637237422s^{26} + 742206852421449807276s^{25} + 3061285798160471289822s^{24} + 10391895636644586020112s^{23} + 29588363727735612069036s^{22} + 71651404108146138897096s^{21} + 149117620073027461547436s^{20} + 268806705178958727187248s^{19} + 422185992215286625454736s^{18} + 580191752386498986323712s^{17} + 699694724310231624064272s^{16} + 741757261801656111225984s^{15} + 691666247103583662351612s^{14} + 567033087779716710023352s^{13} + 408047131644656580559296s^{12} + 257036644794565634383008s^{11} + 141145264141826804073576s^{10} + 67178460532288884909516s^9 + 27499663057942951141041s^8 + 9582491278489242855672s^7 + 2803365139419823150782s^6 + 675682953313836552876s^5 + 130659498671205647052s^4 + 19489563056909654496s^3 + 2105305456045150908s^2 + 146594486051390088s + 4941387170271576)$ |

$$\Delta_2$$

$$-36s^4(2s+3)^4(5s+6)^4(1881888621495012747598719782641000000s^{66}$$
$$+11719822793673389354852048917721870400s^{65}$$
$$+3590340595158437649151724049466274089920s^{64}$$
$$+721208857126645251632241909002396530195200s^{63}$$
$$+1068391320253687840934482144561140800899360s^{62}$$
$$+12446539307453434624006123411233664305561120s^{61}$$
$$+118745312725867591570488714252737764196464800s^{60}$$
$$+953971746397808491174002179252908607975430720s^{59}$$
$$+6586014211961623946750060924092706557833870960s^{58}$$
$$+39680223557336960917652631305424892564932972960s^{57}$$
$$+211171821578745369801696554651560917442769203800s^{56}$$
$$+1002346955322263064366492155231201633504320407840s^{55}$$
$$+4277235454084557392774406702392019604985274451680s^{54}$$
$$+16516932048985831100019838316753837038388031112800s^{53}$$
$$+580388130512851592806447275988950390613484071978800s^{52}$$
$$+18645262399111732404927179918528703191829668776944s^{51}$$
$$+549821609239708629071558539871576928280510022129468s^{50}$$
$$+14934032713121190950602119161930327947606930700148000s^{49}$$
$$+374738908504925756556865067810806785110272348870998s^{48}$$
$$+8709591713633887671245551839384745486844330327339760s^{47}$$
$$+187912542310567752353269511359111865325103054321965000s^{46}$$
$$+3770893425502604495219029177077658520798571516698604s^{45}$$
$$+70500152217841744907941083882518130732596436630910080s^{44}$$
$$+1229751697814540088737430125405972391309535236404816s^{43}$$
$$+200381644841642366592891584643332332297986145636955800s^{42}$$
$$+30532204930658301922179960307897986850960973110401060s^{41}$$
$$+4353974016564230774744937671615227473899797195238128800s^{40}$$
$$+5814794272234047468845528446358546692682944712469086400s^{39}$$
$$+72765609819670245981642226229421333542816165300786478s^{38}$$
$$+85352226271944117403522054472274990615987412077658020s^{37}$$
$$+93861185123580180986528832241395054124125210332787978s^{36}$$
$$+96773676228707454794185949836280770022007026377891360s^{35}$$
$$+93535505340566829835280969592461902479874869737070324s^{34}$$
$$+84726573070331222224010481384087646174794999517275720s^{33}$$
$$+71892923025396467242385567135204296459367606931171072s^{32}$$
$$+57107505385821670765861094421169358296399126222638720s^{31}$$
$$+42429475579369200294980396815322839839865021711481712s^{30}$$
$$+29453402248671311370918975586371950400100875859184424s^{29}$$
$$+19076887418815475591391225344828673331985433949859745s^{28}$$
$$+11509630902613742430056710113025979946368929363591560s^{27}$$
$$+64553145705892629354259479953553982198410836194602750s^{26}$$
$$+33574325610610206035803827859225859005363963939310862s^{25}$$
$$+1614483581569691108434974127018929362203153022365980s^{24}$$
$$+7151843921239467644322741543961403561291367742781520s^{23}$$
$$+29056159370788241154971270303899134550457836920080s^{22}$$
$$+1076866147250386418961409060935085408978065543348840s^{21}$$
$$+3617512376782787657128716857177209606664184218971500s^{20}$$
$$+109362954136214178401472446585845791272071743893520s^{19}$$
$$+2956231446951945885452453177600650222360031507853s^{18}$$
$$+714328315495087334767452511829810830341704532294s^{17}$$
$$+157862256397341201315796262528952028945210160178s^{16}$$
$$+3466918481540893224348429068849868462291600055200s^{15}$$
$$+8792208859688249070338942374434094569299968434s^{14}$$
$$+27289589134979046796056293043482830676621491560s^{13}$$
$$+90851447731067951899123953408477649856954913000s^{12}$$
$$+28199738465157182443141692544401581810169660000s^{11}$$
$$+7659571256132994444113894245418887273849699200s^{10}$$

$$+17823235368805916505433544016195415996681584 s^9$$
$$+35225074079403593877757698013822148284205 16 s^8$$
$$+58678200506460800083204347410791939551955 2 s^7$$
$$+81484523152763229188477555846803640661672 s^6$$
$$+92756899932438870538994638249892996334 4 s^5$$
$$+84380816732192305737995412988161263409 6 s^4$$
$$+58988087589412129623633466476137297856 s^3$$
$$+2972581287433071568118115850779633072 s^2$$
$$+95923391203691628771401672158154016 s$$
$$+14833514103663653933721908 06569392)$$

The last lemma presents some properties for the population-level SGPO and GRPO dynamics.

**Lemma C.4.** *Under the assumptions from Theorem 3.3, the following statements hold true,*

*(i) $p_{SGPO}^{(k)}, q_{SGPO}^{(k)}, p_{GRPO}^{(k)}, q_{GRPO}^{(k)} \in (0,1)$ for all $k \geq 0$.*

*(ii) $p_{SGPO}^{(k)}, q_{SGPO}^{(k)}, p_{GRPO}^{(k)}, q_{GRPO}^{(k)}$ are strictly increasing in $k$ and lie in $(\frac{1}{2}, 1)$ for all $k \geq 1$.*

*(iii) $p_{SGPO}^{(k)} > q_{SGPO}^{(k)}$ for all $k \geq 1$.*

*Proof.* We first rewrite the update rule in Eq. (3) as follows,

$$p_{\text{SGPO}}^{(k+1)} = p_{\text{SGPO}}^{(k)} \frac{e^{\Delta_{\text{SGPO},p}^{(k)}}}{1 - p_{\text{SGPO}}^{(k)} + p_{\text{SGPO}}^{(k)} e^{\Delta_{\text{SGPO},p}^{(k)}}}, \quad \text{where} \quad \Delta_{\text{SGPO},p}^{(k)} = p_{\text{SGPO}}^{(k)}(1 - p_{\text{SGPO}}^{(k)}),$$

$$q_{\text{SGPO}}^{(k+1)} = q_{\text{SGPO}}^{(k)} \frac{e^{\Delta_{\text{SGPO},q}^{(k)}}}{1 - q_{\text{SGPO}}^{(k)} + q_{\text{SGPO}}^{(k)} e^{\Delta_{\text{SGPO},q}^{(k)}}}, \quad \text{where} \quad \Delta_{\text{SGPO},q}^{(k)} = (p_{\text{SGPO}}^{(k)})^2 q_{\text{SGPO}}^{(k)}(1 - q_{\text{SGPO}}^{(k)}),$$

$$p_{\text{GRPO}}^{(k+1)} = p_{\text{GRPO}}^{(k)} \frac{e^{\Delta_{\text{GRPO},p}^{(k)}}}{1 - p_{\text{GRPO}}^{(k)} + p_{\text{GRPO}}^{(k)} e^{\Delta_{\text{GRPO},p}^{(k)}}}, \quad \text{where} \quad \Delta_{\text{GRPO},p}^{(k)} = p_{\text{GRPO}}^{(k)}(1 - p_{\text{GRPO}}^{(k)}) q_{\text{GRPO}}^{(k)},$$

$$q_{\text{GRPO}}^{(k+1)} = q_{\text{GRPO}}^{(k)} \frac{e^{\Delta_{\text{GRPO},q}^{(k)}}}{1 - q_{\text{GRPO}}^{(k)} + q_{\text{GRPO}}^{(k)} e^{\Delta_{\text{GRPO},q}^{(k)}}}, \quad \text{where} \quad \Delta_{\text{GRPO},q}^{(k)} = p_{\text{GRPO}}^{(k)} q_{\text{GRPO}}^{(k)}(1 - q_{\text{GRPO}}^{(k)}).$$

First of all, the uniform initialization yields the desired result for $k = 0$. Suppose $p_{\text{SGPO}}^{(k)} \in (0,1)$ for some $k \geq 0$. Then, we have

$$1 - p_{\text{SGPO}}^{(k)} + p_{\text{SGPO}}^{(k)} e^{\Delta_{\text{SGPO},p}^{(k)}} > p_{\text{SGPO}}^{(k)} e^{\Delta_{\text{SGPO},p}^{(k)}} > 0,$$

which implies $p_{\text{SGPO}}^{(k+1)} \in (0,1)$. By induction, we have $p_{\text{SGPO}}^{(k)} \in (0,1)$ for all $k \geq 0$. Similarly, we can show that $q_{\text{SGPO}}^{(k)}, p_{\text{GRPO}}^{(k)}, q_{\text{GRPO}}^{(k)} \in (0,1)$ for all $k \geq 0$.

Furthermore, we have $\Delta_{\text{SGPO},p}^{(k)} > 0$ since $p_{\text{SGPO}}^{(k)} \in (0,1)$. This implies

$$\frac{p_{\text{SGPO}}^{(k+1)}}{p_{\text{SGPO}}^{(k)}} = \frac{1}{(1 - p_{\text{SGPO}}^{(k)}) e^{-\Delta_{\text{SGPO},p}^{(k)}} + p_{\text{SGPO}}^{(k)}} > \frac{1}{1 - p_{\text{SGPO}}^{(k)} + p_{\text{SGPO}}^{(k)}} = 1.$$

Since $p_{\text{SGPO}}^{(0)} = \frac{1}{2}$, we have $p_{\text{SGPO}}^{(k)} \in (\frac{1}{2}, 1)$ for all $k \geq 1$. Similarly, we can show that $q_{\text{SGPO}}^{(k)}, p_{\text{GRPO}}^{(k)}, q_{\text{GRPO}}^{(k)}$ are strictly increasing and lie in $(\frac{1}{2}, 1)$.

Finally, we have $p_{\text{SGPO}}^{(0)} \geq q_{\text{SGPO}}^{(0)}$. Thus, it suffices to show that $p_{\text{SGPO}}^{(k)} \geq q_{\text{SGPO}}^{(k)}$ implies $p_{\text{SGPO}}^{(k+1)} > q_{\text{SGPO}}^{(k+1)}$ for all $k \geq 0$. Indeed, Lemma C.1(i) and $p_{\text{SGPO}}^{(k)} \geq q_{\text{SGPO}}^{(k)}$ yield

$$p_{\text{SGPO}}^{(k+1)} = \exp(f_{11}(p_{\text{SGPO}}^{(k)})) \geq \exp(f_{11}(q_{\text{SGPO}}^{(k)})) = \exp(\log(q_{\text{SGPO}}^{(k)}) + h_{q_{\text{SGPO}}^{(k)}}(q_{\text{SGPO}}^{(k)}(1 - q_{\text{SGPO}}^{(k)}))).$$

Then, Lemma C.1(ii) and $p_{\text{SGPO}}^{(k)} \in (0,1)$ yield

$$\exp(\log(q_{\text{SGPO}}^{(k)}) + h_{q_{\text{SGPO}}^{(k)}}(q_{\text{SGPO}}^{(k)}(1 - q_{\text{SGPO}}^{(k)}))) > \exp(\log q_{\text{SGPO}}^{(k)} + h_{q_{\text{SGPO}}^{(k)}}((p_{\text{SGPO}}^{(k)})^2 q_{\text{SGPO}}^{(k)}(1 - q_{\text{SGPO}}^{(k)}))).$$

In addition, we have

$$q_{\text{SGPO}}^{(k+1)} = \exp(f_{12}(p_{\text{SGPO}}^{(k)}, q_{\text{SGPO}}^{(k)})) = \exp(\log q_{\text{SGPO}}^{(k)} + h_{q_{\text{SGPO}}^{(k)}}((p_{\text{SGPO}}^{(k)})^2 q_{\text{SGPO}}^{(k)}(1 - q_{\text{SGPO}}^{(k)}))).$$

Putting these pieces together yields $p_{\text{SGPO}}^{(k+1)} > q_{\text{SGPO}}^{(k+1)}$. □

C.3   PROOF OF THEOREM 3.3

To show (i), recall that the sequence $(p_{\text{SGPO}}^{(k)})_{k\in\mathbb{N}}$ is strictly increasing and bounded in $(0,1)$ from Lemmas C.4(i) and C.4(ii), so it converge to some value $c \in (0,1]$. Take limit as $k \to \infty$:

$$1 = \lim_{k\to\infty} \frac{p_{\text{SGPO}}^{(k+1)}}{p_{\text{SGPO}}^{(k)}} = \lim_{k\to\infty} \frac{1}{(1-p_{\text{SGPO}}^{(k)})e^{-\Delta_{\text{SGPO},p}^{(k)}}+p_{\text{SGPO}}^{(k)}} = \frac{1}{(1-c)e^{-c(1-c)+c}}.$$

Using the simple Taylor lower bound $e^{-x} \geq 1 - x$, we have

$$1 = \frac{1}{(1-c)e^{-c(1-c)}+c} \geq \frac{1}{(1-c)(1-c(1-c))+c} \implies (c-1)^2 \leq 0 \implies c = 1.$$

This shows $p_{\text{SGPO}}^{(k)} \to 1$ as $k \to \infty$. Similarly, we can show $q_{\text{GRPO}}^{(k)}, p_{\text{SGPO}}^{(k)}, q_{\text{SGPO}}^{(k)} \to 1$ as $k \to \infty$.

To show (ii), consider the base case:

$$p_{\text{SGPO}}^{(1)} = \exp(f_{11}(p_{\text{SGPO}}^{(0)})) = \exp(\log p_{\text{SGPO}}^{(0)} + h_{p_{\text{SGPO}}^{(0)}}(p_{\text{SGPO}}^{(0)}(1 - p_{\text{SGPO}}^{(0)})))$$

$$= \exp(\log p_{\text{GRPO}}^{(0)} + h_{p_{\text{GRPO}}^{(0)}}(p_{\text{GRPO}}^{(0)}(1 - p_{\text{GRPO}}^{(0)})))$$

$$> \exp(\log p_{\text{GRPO}}^{(0)} + h_{p_{\text{GRPO}}^{(0)}}(p_{\text{GRPO}}^{(0)}(1 - p_{\text{GRPO}}^{(0)})q_{\text{GRPO}}^{(0)})) = \exp(f_{21}(p_{\text{GRPO}}^{(0)}, q_{\text{GRPO}}^{(0)})) = p_{\text{GRPO}}^{(1)},$$

where the inequality follows from Lemma C.1(ii). Thus, we use induction and assume $p_{\text{SGPO}}^{(k)} > p_{\text{GRPO}}^{(k)}$ for some $k \geq 1$. Then we have,

$$p_{\text{SGPO}}^{(k+1)} = \exp(f_{11}(p_{\text{SGPO}}^{(k)})) > \exp(f_{11}(p_{\text{GRPO}}^{(k)})) = \exp(\log p_{\text{GRPO}}^{(k)} + h_{p_{\text{GRPO}}^{(k)}}(p_{\text{GRPO}}^{(k)}(1 - p_{\text{GRPO}}^{(k)})))$$

$$> \exp(\log p_{\text{GRPO}}^{(k)} + h_{p_{\text{GRPO}}^{(k)}}(p_{\text{GRPO}}^{(k)}(1 - p_{\text{GRPO}}^{(k)})q_{\text{GRPO}}^{(k)})) = \exp(f_{21}(p_{\text{GRPO}}^{(k)}, q_{\text{GRPO}}^{(k)})) = p_{\text{GRPO}}^{(k+1)},$$

where the first inequality uses Lemma C.1(i) and the second one uses Lemma C.1(ii). Thus, $p_{\text{SGPO}}^{(k+1)} > p_{\text{GRPO}}^{(k+1)}$ and induction completes. We have proved that $p_{\text{SGPO}}^{(k)} > p_{\text{GRPO}}^{(k)}$ for all $k \geq 1$.

To show (iii), first notice that we can show $p_{\text{GRPO}}^{(k)} = q_{\text{GRPO}}^{(k)}$ for all $k \geq 0$ by induction. The base case is trivial by initialization. Suppose $p_{\text{GRPO}}^{(k)} = q_{\text{GRPO}}^{(k)}$ for some $k \geq 0$, then by noticing that $f_{21}(p,p) = f_{22}(p,p)$, we have

$$p_{\text{GRPO}}^{(k+1)} = \exp(f_{21}(p_{\text{GRPO}}^{(k)}, q_{\text{GRPO}}^{(k)})) = \exp(f_{21}(p_{\text{GRPO}}^{(k)}, p_{\text{GRPO}}^{(k)}))$$

$$= \exp(f_{22}(p_{\text{GRPO}}^{(k)}, p_{\text{GRPO}}^{(k)})) = \exp(f_{22}(p_{\text{GRPO}}^{(k)}, q_{\text{GRPO}}^{(k)})) = q_{\text{GRPO}}^{(k+1)}.$$

Thus, by induction, $p_{\text{GRPO}}^{(k)} = q_{\text{GRPO}}^{(k)}$ for all $k \geq 0$. Now, we can reduce the update rule of $p_{\text{GRPO}}^{(k)}$ as

$$p_{\text{GRPO}}^{(k+1)} = \frac{1}{(1/p_{\text{GRPO}}^{(k)}-1)\exp(-(p_{\text{GRPO}}^{(k)})^2(1-p_{\text{GRPO}}^{(k)}))+1}.$$

Also recall the update rule of $p_{\text{SGPO}}^{(k)}$ and $q_{\text{SGPO}}^{(k)}$:

$$\begin{aligned} p_{\text{SGPO}}^{(k+1)} &= \frac{1}{(1/p_{\text{SGPO}}^{(k)}-1)\exp(-p_{\text{SGPO}}^{(k)}(1-p_{\text{SGPO}}^{(k)}))+1} \\ q_{\text{SGPO}}^{(k+1)} &= \frac{1}{(1/q_{\text{SGPO}}^{(k)}-1)\exp(-(p_{\text{SGPO}}^{(k)})^2 q_{\text{SGPO}}^{(k)}(1-q_{\text{SGPO}}^{(k)}))+1} \end{aligned},$$

and it suffices to show $p_{\text{SGPO}}^{(k)} q_{\text{SGPO}}^{(k)} > (p_{\text{GRPO}}^{(k)})^2$ for all $k \geq 1$. We prove by induction. For the base case,

$$\sqrt{p_{\text{SGPO}}^{(1)} q_{\text{SGPO}}^{(1)}} = \sqrt{\frac{1}{1+(1/2)e^{-1/4}} \cdot \frac{1}{1+(1/2)e^{-1/16}}} > \frac{1}{1+(1/2)e^{-1/8}} = p_{\text{GRPO}}^{(1)}.$$

The above inequality holds true since Lemma C.1(iv) implies

$$2\log(1 + (1/2)e^{-1/8}) > \log(1 + (1/2)e^{-1/4}) + \log(1 + (1/2)e^{-1/16}),$$

It remains to show that $p_{\text{SGPO}}^{(k)} q_{\text{SGPO}}^{(k)} > (p_{\text{GRPO}}^{(k)})^2$ implies $p_{\text{SGPO}}^{(k+1)} q_{\text{SGPO}}^{(k+1)} > (p_{\text{GRPO}}^{(k+1)})^2$ for $k \geq 1$. By Lemma C.4(iii), we know $p_{\text{SGPO}}^{(k)} > q_{\text{SGPO}}^{(k)}$ for all $k \geq 1$. Thus, Lemma C.2 implies that

$$p_{\text{SGPO}}^{(k+1)} q_{\text{SGPO}}^{(k+1)} = \frac{1}{A(p_{\text{SGPO}}^{(k)})B(p_{\text{SGPO}}^{(k)}, q_{\text{SGPO}}^{(k)})} > \frac{1}{C\left(\sqrt{p_{\text{SGPO}}^{(k)} q_{\text{SGPO}}^{(k)}}\right)^2}.$$

Using Lemma C.1(iii), we complete the induction by applying our induction hypothesis:

$$\frac{1}{C\left(\sqrt{p_{\text{SGPO}}^{(k)} q_{\text{SGPO}}^{(k)}}\right)^2} = \exp\left(f_{21}\left(\sqrt{p_{\text{SGPO}}^{(k)} q_{\text{SGPO}}^{(k)}}, \sqrt{p_{\text{SGPO}}^{(k)} q_{\text{SGPO}}^{(k)}}\right)\right) > \exp(f_{21}(p_{\text{GRPO}}^{(k)}, p_{\text{GRPO}}^{(k)})) = (p_{\text{GRPO}}^{(k+1)})^2.$$

This completes the proof.

### C.4 EMPIRICAL EVALUATIONS

We conduct simple simulations to compare SGPO and GRPO under the stylized model and plot the resulting learning curves in Figure 4. The left panel shows the probability of selecting the "good" action in the first step at iteration $k$ (i.e., $p^{(k)}_{\text{SGPO}}$ vs. $p^{(k)}_{\text{GRPO}}$), while the right panel shows the probability of learning the optimal policy (i.e., $p^{(k)}_{\text{SGPO}}q^{(k)}_{\text{SGPO}}$ vs. $p^{(k)}_{\text{GRPO}}q^{(k)}_{\text{GRPO}}$). The results align with the predictions of Theorem 3.3, demonstrating that the likelihood of learning the optimal policy under SGPO consistently exceeds that of GRPO across training.

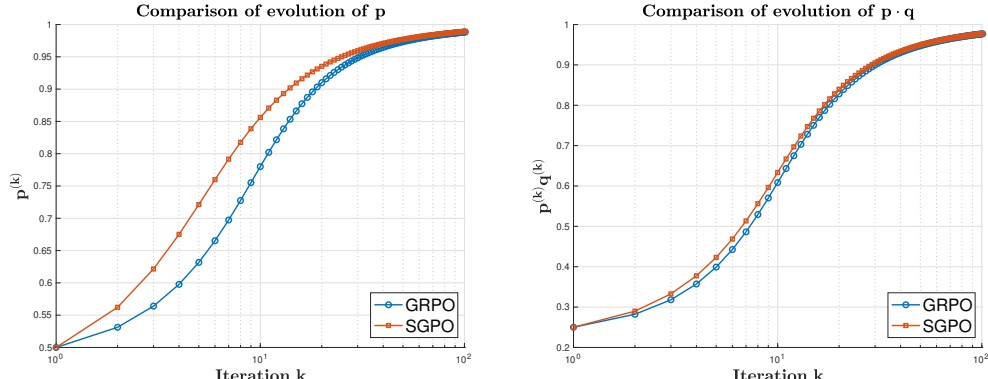

Figure 4: Learning dynamics of GRPO and SGPO in the simplified setting.

## D THE USE OF LARGE LANGUAGE MODELS (LLMS)

In addition to serving as the step-wise judge model, we also use LLMs to aid and polish the writing. Specifically, we employ them for grammar and style checking to improve the readability of this work.

