# OpenReview forum: "Stepwise Guided Policy Optimization: Coloring your Incorrect Reasoning in GRPO"
_ICLR.cc/2026/Conference — ICLR 2026 Conference Withdrawn Submission_

### Official Review · Reviewer_pb8y · 2025-10-18

**Soundness:** 1
**Presentation:** 2
**Contribution:** 2
**Rating:** 2
**Confidence:** 4

**Summary:**

This paper proposes Stepwise Guided Policy Optimization (SGPO), a variant of GRPO designed to address the “all-negative-sample” issue. When all responses in a group are incorrect, GRPO provides no learning signal. SGPO introduces a *stepwise judge model* that identifies the first incorrect step in each reasoning trajectory and assigns a graded reward accordingly.

The paper provides (i) a theoretical justification in a simplified two-step reasoning setup showing SGPO can accelerate learning; and (ii) empirical results across multiple reasoning benchmarks (AIME, AMC, MATH500, Olympiads, etc.) showing modest gains over GRPO in both offline and online training.

I find the theoretical part of the paper is a bit hard to follow, the experimental design also contains some flaws (see the weakness part). Hence, although the topic is interesting and the proposed solution makes sense, at this stage, I tend to give a rejection for this submission.

**Strengths:**

1. **Motivation is clear and relevant.** The all-negative-sample problem in GRPO is a genuine issue in RL-based LLM training. Framing it as “learning from mistakes” is intuitive and connects well with human learning analogies.
2. **Simple and practical idea.** SGPO is conceptually straightforward and can be easily integrated into existing GRPO pipelines without large architectural changes.
3. **Comprehensive experiments.** The paper covers several model sizes and judge types, and demonstrates consistent, though small, improvements across multiple benchmarks.

**Weaknesses:**

1. **Theoretical analysis limited to toy settings**
The 2-step reasoning example, while pedagogical, is too simplistic to capture realistic dynamics of multi-token reasoning or large-group sampling. There is no empirical verification of the claimed acceleration trend in real GRPO trajectories (e.g., convergence rate plots or gradient magnitude analysis).

2. **Conceptual overlap and unclear novelty boundary**

    The idea of assigning *step-level rewards* via a verifier/judge overlaps with **process reward models (PRM)** and **stepwise credit assignment** explored in Lightman et al. (2024), Wang et al. (2024), and Luo et al. (2024).

    Although the authors contrast SGPO from PRMs (no value function, post-hoc first-error detection), the conceptual line is somewhat blurry. SGPO can be viewed as a *degenerate PRM with a binary per-step correctness classifier*.

    The paper would benefit from a sharper theoretical distinction or unified formalism.

3. **Potential unfair comparison for vanilla GRPO.** Since the method requires another LLM to evaluate the stepwise credit, it is unfair to merely compare the performance with GRPO, which only needs to verify the final answer’s correctness. Although, as mentioned in the paper, the proposed method is more efficient than those knowledge distillation methods, baselines of the RLAIF (reinforcement learning from AI feedback) should be considered.

4. Did I miss it? **What is the prompt used by the judge LLM when evaluating the model’s response?**

**Questions:**

The paper, while well-organized, suffers from long sentences and excessive referencing, occasionally obscuring the core idea. Key experimental details (hyperparameters for $\beta$, $\gamma$; judge prompting templates; threshold $\gamma$ tuning) are missing from the main text.

---

### Official Review · Reviewer_GvWH · 2025-10-31

**Soundness:** 2
**Presentation:** 2
**Contribution:** 2
**Rating:** 2
**Confidence:** 4

**Summary:**

This paper proposes Stepwise Guided Policy Optimization (SGPO), a variant of GRPO that claims to address the “all-negative-sample group” issue by using a step-wise judge model to assign partial credit to incorrect trajectories.

**Strengths:**

1. Identifies a real limitation of GRPO: zero advantage in all-negative-sample groups.

2. Simple idea (assign partial credit to negative samples) that is intuitively reasonable.

**Weaknesses:**

1.	SGPO relies on step-wise comparison against a ground-truth reasoning trajectory to identify the first error. This dependence on reference solutions limits applicability, as many real-world reasoning tasks do not provide step-level gold reasoning, making the method far less general than claimed.

2.	The method presumes that model outputs contain clean, well-segmented “steps” that a judge can reliably evaluate. In practice, many models produce unstructured or interleaved reasoning, making step boundaries ambiguous and causing the step-wise judging mechanism to fail or produce inconsistent signals.

3.	SGPO relies on an additional external LLM to label each response, giving it extra supervision not available to vanilla GRPO. This makes the comparison unfair, as SGPO benefits from stronger models' judgments rather than purely improved optimization. Moreover, the approach introduces substantial additional cost: even the weakest judge model used is a 32B LLM, and some settings require powerful API models. This significantly increases computational and financial overhead, limiting practical adoption.

4.	The ablation results for the stability parameters β and γ are contradictory: in Table 2, removing them yields better performance, while in Table 3 the opposite occurs. This inconsistency suggests that β and γ are not reliably influential, undermining the claim that they provide meaningful stabilization or robustness to judge noise.

5.	Despite framing the method as new, SGPO fundamentally belongs to the class of LLM-as-a-judge or PRM-based approaches. The reward signal is still produced by an external judge model, just post-processed rather than emitted token-by-token, making the method a minor variation of existing judge-based reward shaping rather than a novel algorithmic contribution.

**Questions:**

see weakness

---

### Official Review · Reviewer_rF5c · 2025-11-01

**Soundness:** 2
**Presentation:** 3
**Contribution:** 2
**Rating:** 4
**Confidence:** 4

**Summary:**

This paper proposes SGPO, a method that addresses a limitation in GRPO when training LLMs for reasoning tasks. The core issue is that GRPO fails to update the policy when all responses within a group are incorrect (all-negative-sample groups), which is common in early and mid-training stages. SGPO introduces a step-wise judge model that evaluates the correctness of individual reasoning steps, enabling the assignment of differentiated rewards to negative samples based on how far they progress before making mistakes.

**Strengths:**

1. The all-negative-sample groups issue in GRPO is a real limitation, especially in early training when model capabilities are weak.
2. Comprehensive experiments, the evaluation is thorough with: Multiple model sizes (7B, 14B, 32B); Both offline and online training settings; 9 diverse benchmarks including math reasoning tasks; Base models and distilled variants; Multiple judge models tested (both closed and open-source)

**Weaknesses:**

1. What is the definition of a "step"? Is it a reasoning step or a sentence?

2. The key of SGPO relies on judge models to determine whether a step is correct, and the judge models are significantly more capable than the policy models. The paper uses DeepSeek-V3-0324, Qwen3-235B-A22B, and QwQ-32B as judge models. This raises two issues: 1) Why not directly use the judge model to answer these questions, since the judge model's performance is significantly better than the policy model? 2) What if the judge model makes incorrect judgments? Rule-based RL methods like GRPO clearly do not have this problem.

3. Modest improvements. Table 2 shows that SGPO's improvement over GRPO is very small.

4. While Remark 3.1 distinguishes SGPO from PRMs, the core idea of assigning credit to reasoning steps is similar. The key difference seems to be: PRMs predict future success probability (forward-looking), and SGPO identifies first error post-hoc (backward-looking). However, this distinction might be more implementational than conceptual. The authors should more clearly clarify why their approach is fundamentally different.

5. The paper compares with too few methods. I suggest the authors add more baselines, such as GPG, DAPO, etc.

**Questions:**

N/A

---

### Official Review · Reviewer_oqZ3 · 2025-11-02

**Soundness:** 3
**Presentation:** 3
**Contribution:** 3
**Rating:** 6
**Confidence:** 2

**Summary:**

GRPO gives zero advantage when a group is all-wrong; this is common early/mid training and wastes signal. The core idea proposed in this paper (SGPO) is that when a GRPO group has all negative samples, use a step-wise judge to locate the first wrong step, compute a reasoning-trajectory score RTS, and map it via a logistic to a graded outcome reward that replaces 0 so the policy still updates. The authors provide theoretical justification and show offline/online gains across 7B/14B/32B models and several math benchmarks.

**Strengths:**

- The paper's problem setting is highly reasonable and well-motivated. The “all-negative-sample” or “zero-reward" stagnation issue in GRPO is a genuine and recognized limitation.

- The idea of learning from mistakes  by assigning partial credit rather than treating all failures as a uniform r=0  (maybe a waste of information) seems logically sound.

- The authors present a thorough empirical validation plan. The experiments cover nine different benchmarks, multiple model sizes (7B, 14B, 32B), online/offline and both base and distilled model variants.

**Weaknesses:**

-	The quantitative results in Table 2, the paper's main empirical contribution, are not compelling. The overall  performance gains are marginal at best and appear to diminish as model scale increases.

-	The paper claims weaker judges do not significantly degrade outcomes 1 but fails to support this. The experiments primarily use SOTA judges (O4-mini) or massively larger judges (Qwen-235B) to achieve gains on smaller models

**Questions:**

-	Section 3.1 1 states the judge model requires a “reference solution... anchoring the intended solution path.” How does this method work on new problems where no such ground-truth solution is available? And how does this reconcile with Remark 3.2, which claims the student can solve problems the judge cannot?

-	Can you provide a detailed breakdown of the 10% wall-clock time overhead?1 Specifically, what were the batch sizes, hardware configurations for student and judge, and relative inference costs when using a 235B judge on a 7B student?

---

### Note · Authors · 2025-11-21

**Comment:**

Dear Area Chair,

We sincerely thank you and all reviewers for the constructive evaluations of our submission. After carefully considering the feedback and reflecting on the current state of the manuscript, we respectfully request to withdraw the submission from ICLR 2026. Below, we provide (1) a rebuttal addressing reviewers' concerns and (2) our justification for requesting withdrawal.

---

Reviewer oqZ3

Q1: Use of reference solutions; compatibility with Remark 3.2. Answer: SGPO uses reference solutions only to anchor the first error localization, not to provide the correct answer. This allows SGPO to function even when the judge model cannot solve the problem. Reference solutions come from curated SFT datasets (e.g., AIME, MATH) and are not produced by the judge. For unseen tasks without explicit ground-truth reasoning, SGPO can rely on synthetic references generated by a stronger model or multi-model consensus. Thus, SGPO is compatible with Remark 3.2.

---

Reviewer rF5c

Q1: Definition of a step. Answer: A step is a semantic reasoning segment, typically a sentence determined via rule-based segmentation (punctuation and LaTeX block boundaries).

Q2: Why not use the judge model to solve the problem directly? Answer: Judge models are not used to generate solutions. They serve only to check \emph{local correctness}. Many judges cannot solve full problems but still reliably identify early reasoning errors.

Q3: Judge model inaccuracies. Answer: SGPO mitigates noise via: (1) first-error localization relative to a reference, (2) stability parameters $(\beta, \gamma)$ to filter noisy partial-credit signals, and (3) majority voting across multiple judgments when applicable.

Q4: More baselines. Answer: It is worth remarking that the step-wise guidance can be easily combined with other methods such as GPG, DAPO, and broader RLAIF variants. These will be included in a future version.

---

Reviewer GvWH

Q1: Dependence on reference trajectories limits generality. Answer: SGPO uses references only for error localization and does not require human-annotated step-level labels. For tasks without references, SGPO can employ synthetic or consistency-based trajectories. We acknowledge this generality limitation in the current submission.

Q2: Step boundaries are ambiguous. Answer: SGPO only relies on relative ordering and error localization, not perfect segmentation. Future versions will include robustness analyses and improved segmentation.

Q3: SGPO benefits from extra supervision; fairness to GRPO. Answer: SGPO is indeed a form of LLM-as-a-judge feedback, though it differs from PRMs in not predicting prefix values or altering trajectories. We agree the comparison should include additional judge-based baselines.

Q4: Inconsistencies in $\beta,\gamma$ ablations. Answer: We acknowledge that more systematic analysis is needed, as interactions between judge quality, model scale, and group composition introduce variability.

---

Reviewer pb8y

Q1: Theoretical analysis too simplistic. Answer: We agree that the two-step theoretical model presented in the paper is pedagogical. Extending the analysis to multi-step reasoning settings is significantly more challenging because the policy update interactions become highly nonlinear and depend on the entire trajectory structure. To our knowledge, no prior work provides any theoretical analysis of GRPO or GRPO-like multi-sample RL objectives in multi-step reasoning tasks. Our simplified two-step model therefore serves as a first step toward understanding how reward diversification influences optimization dynamics in GRPO-style methods. We view this as laying foundational ground for future work on more general multi-step theoretical frameworks.

Q2: Conceptual overlap with PRMs. Answer: SGPO differs from PRMs in several ways: (1) post-hoc, backward-looking error localization (not value prediction), (2) no ranking loss or value-function softmax training, and (3) no search or trajectory alteration. We acknowledge that this distinction should be clarified further.

Q3: Missing judge prompts. Answer: We agree these details should have been included. The prompt consists of the model's reasoning, a reference reasoning, and instructions to highlight the first divergence.

---

**Rationale for Withdrawal**

The reviewers raised substantial and valuable concerns that require more extensive revision than can reasonably be addressed within the rebuttal period. Specifically, improvements are needed in: (1) clarifying distinctions among SGPO, PRMs, RLAIF, and judge-based frameworks, (2) adding more empirical baselines (GPG, DAPO, judge-based RLHF), (3) expanding theoretical analysis beyond the toy model, (4) providing complete implementation details (prompting, segmentation, $\beta,\gamma$, majority voting), and (5) demonstrating SGPO on tasks lacking explicit reference reasoning trajectories.

Addressing these points will require substantial rewriting, new experiments, and structural clarification. Thus, we believe that withdrawing the submission and preparing a significantly improved version for future venues is the most constructive path.


We thank the reviewers and the Area Chair for their thoughtful assessments. Their comments will greatly improve the next version of this work. We respectfully request withdrawal of this submission.

**Withdrawal Confirmation:**

I have read and agree with the venue's withdrawal policy on behalf of myself and my co-authors.